# Estimating informativeness of samples with Smooth Unique Information

**Hrayr Harutyunyan**[1,2][*]**, Alessandro Achille**[1]**, Giovanni Paolini**[1]**, Orchid Majumder**[1]**,
Avinash Ravichandran**[1]**, Rahul Bhotika**[1]**, Stefano Soatto**[1]
[1] Amazon Web Services, [2] USC Information Sciences Institute
`hrayrhar@usc.edu`, {`aachille, paoling, orchid`}`@amazon.com`,
{`ravinash, bhotikar, soattos`}`@amazon.com`

## Abstract

We define a notion of information that an individual sample provides to the training of a neural network, and we specialize it to measure both how much a sample informs the final weights and how much it informs the function computed by the weights. Though related, we show that these quantities have a qualitatively different behavior. We give efficient approximations of these quantities using a linearized network and demonstrate empirically that the approximation is accurate for real-world architectures, such as pre-trained ResNets. We apply these measures to several problems, such as dataset summarization, analysis of under-sampled classes, comparison of informativeness of different data sources, and detection of adversarial and corrupted examples. Our work generalizes existing frameworks but enjoys better computational properties for heavily over-parametrized models, which makes it possible to apply it to real-world networks.

## 1 Introduction

Training a deep neural network (DNN) entails extracting information from samples in a dataset and storing it in the weights of the network, so that it may be used in future inference or prediction. But how much information does a *particular* sample contribute to the trained model? The answer can be used to provide strong generalization bounds (if no information is used, the network is not memorizing the sample), privacy bounds (how much information the network can leak about a particular sample), and enable better interpretation of the training process and its outcome. To determine the information content of samples, we need to define and compute information. In the classical sense, information is a property of random variables, which may be degenerate for the deterministic process of computing the output of a trained DNN in response to a given input (inference). So, even posing the problem presents some technical challenges. But beyond technicalities, how can we know whether a given sample is memorized by the network and, if it is, whether it is used for inference?

We propose a notion of *unique sample information* that, while rooted in information theory, captures some aspects of stability theory and influence functions. Unlike most information-theoretic measures, ours can be approximated efficiently for large networks, especially in the case of transfer learning, which encompasses many real-world applications of deep learning. Our definition can be applied to either "weight space" or "function space." This allows us to study the non-trivial difference between information the weights possess (weight space) and the information the network actually *uses* to make predictions on new samples (function space).

Our method yields a valid notion of information without relying on the randomness of the training algorithm (*e.g.*, stochastic gradient descent, SGD), and works even for deterministic training algorithms. Our main work-horse is a first-order approximation of the network. This approximation is accurate when the network is pre-trained (Mu et al., 2020) — as is common in practical applications — or is randomly initialized but very wide (Lee et al., 2019), and can be used to obtain a closed-form expression of the per-sample information. In addition, our method has better scaling with respect to the number of parameters than most other information measures, which makes it applicable to mas-

---

[*]Work conducted at Amazon Web Services.

sively over-parametrized models such as DNNs. Our information measure can be computed without actually training the network, making it amenable to use in problems like dataset summarization.

We apply our method to remove a large portion of uninformative examples from a training set with minimum impact on the accuracy of the resulting model (dataset summarization). We also apply our method to detect mislabeled samples, which we show carry more unique information.

To summarize, our contributions are (1) We introduce a notion of unique information that a sample contributes to the training of a DNN, both in weight space and in function space, and relate it with the stability of the training algorithm; (2) We provide an efficient method to compute unique information even for large networks using a linear approximation of the DNN, and without having to train a network; (3) We show applications to dataset summarization and analysis. The implementation of the proposed method and the code for reproducing the experiments is available at `https:// github.com/awslabs/aws-cv-unique-information`.

**Prerequisites and Notation.** Consider a dataset of $n$ labeled examples $S = \{z_i\}_{i=1}^n$, where $z_i = (x_i, y_i)$, $x_i \in \mathcal{X}$ and $y_i \in \mathbb{R}^k$ and a neural network model $f_w : \mathcal{X} \mapsto \mathbb{R}^k$ with parameters $w \in \mathbb{R}^d$. Throughout the paper $S_{-i} = \{z_1, \ldots, z_{i-1}, z_{i+1}, \ldots, z_n\}$ denotes the set excluding the $i$-th sample; $f_{w_t}$ is often shortened to $f_t$; the concatenation of all training examples is denoted by $X$; the concatenation of all training labels by $Y \in \mathbb{R}^{nk}$; and the concatenation of all outputs by $f_w(X) \in \mathbb{R}^{nk}$. The loss on the $i$-th example is denoted by $\mathcal{L}_i(w)$ and is equal to $\frac{1}{2}\|f_w(x_i) - y_i\|_2^2$, unless specified otherwise. This choice is useful when dealing with linearized models and is justified by Hui & Belkin (2020), who showed that the mean-squared error (MSE) loss is as effective as cross-entropy for classification tasks. The total loss is $\mathcal{L}(w) = \sum_{i=1}^n \mathcal{L}_i(w) + \frac{\lambda}{2}\|w - w_0\|_2^2$, where $\lambda \geq 0$ is a weight decay regularization coefficient and $w_0$ is the weight initialization point. Note that the regularization term differs from standard weight decay $\|w\|_2^2$ and is more appropriate for linearized neural networks, as it allows us to derive the dynamics analytically (see Sec. F of the appendix). Finally, a (possibly stochastic) training algorithm is denoted with a mapping $A : \mathcal{S} \rightarrow \mathcal{W}$, which maps a training dataset $S$ to classifier weights $W = A(S)$. Since the training algorithm can be stochastic, $W$ is a random variable. The distribution of possible weights $W$ after training with the algorithm $A$ on the dataset $S$ is denoted with $p_A(w \mid S)$. We use several information-theoretic quantities, such as entropy: $H(X) = -\mathbb{E}\big[\log p(x)\big]$, mutual information: $I(X;Y) = H(X) + H(Y) - H(X,Y)$, Kullback–Leibler divergence: $\mathrm{KL}(p(x)\|q(x)) = \mathbb{E}_{x\sim p(x)}\left[\log(p(x)/q(x))\right]$ and their conditional variants (Cover & Thomas, 2006). If $y \in \mathbb{R}^m$ and $x \in \mathbb{R}^n$, then the Jacobian $\frac{\partial y}{\partial x}$ is an $m \times n$ matrix. The gradient $\nabla_x y$ denotes transpose of the Jacobian $\frac{\partial y}{\partial x}$, an $n \times m$ matrix.

## 2 RELATED WORK

Our work is related to information-theoretic stability notions (Bassily et al., 2016; Raginsky et al., 2016; Feldman & Steinke, 2018) that seek to measure the influence of a sample on the output, and to measure generalization. Raginsky et al. (2016) define information stability as $\mathbb{E}_S\left[\frac{1}{n}\sum_{i=1}^n I(W; Z_i \mid S_{-i})\right]$, the expected average amount of unique (Shannon) information that weights have about an example. This, without the expectation over $S$, is also our starting point (eq. 1). Bassily et al. (2016) define KL-stability $\sup_{S,S'} \mathrm{KL}(p_A(w \mid S) \| p_A(w \mid S'))$, where $S$ and $S'$ are datasets that differ by one example, while Feldman & Steinke (2018) define average leave-one-out KL stability as $\sup_S \frac{1}{n}\sum_{i=1}^n \mathrm{KL}(p_A(w \mid S) \| p_A(w \mid S_{-i}))$. The latter closely resembles our definition (eq. 4). Unfortunately, while the weights are continuous, the optimization algorithm (such as SGD) is usually discrete. This generally makes the resulting quantities degenerate (infinite). Most works address this issue by replacing the discrete optimization algorithm with a continuous one, such as stochastic gradient Langevin dynamics (Welling & Teh, 2011) or continuous stochastic differential equations that approximate SGD (Li et al., 2017) in the limit. We aim to avoid such assumptions and give a definition that is directly applicable to real networks trained with standard algorithms. To do this, we apply a smoothing procedure to a standard discrete algorithm. The final result can still be interpreted as a valid bound on Shannon mutual information, but for a slightly modified optimization algorithm. Our definitions relate informativeness of a sample to the notion of algorithmic stability (Bousquet & Elisseeff, 2002; Hardt et al., 2015), where a training algorithm $A$ is called stable if $A(S)$ is close to $A(S')$ when the datasets $S$ and $S'$ differ by only one sample.

To ensure our quantities are well-defined, we apply a smoothing technique which is reminiscent of a soft discretization of weight space. In Section 4, we show that a canonical discretization is obtained using the Fisher information matrix, which relates to classical results of Rissanen (1996) on optimal coding length. It also relates to the use of a post-distribution in Achille et al. (2019), who however use it to estimate the total amount of information in the weights of a network.

We use a first-order approximation (linearization) inspired by the Neural Tangent Kernel (NTK) (Jacot et al., 2018; Lee et al., 2019) to efficiently estimate informativeness of a sample. While NTK predicts that, in the limit of an infinitely wide network, the linearized model is an accurate approximation, we do not observe this on more realistic architectures and datasets. However, we show that, when using pre-trained networks as common in practice, linearization yields an accurate approximation, similarly to what is observed by Mu et al. (2020). Shwartz-Ziv & Alemi (2020) study the total information contained by an ensemble of randomly initialized linearized networks. They notice that, while considering ensembles makes the mutual information finite, it still diverges to infinity as training time goes to infinity. On the other hand, we consider the unique information about a single example, without the neeed for ensembles, by considering smoothed information, which remains bounded for any time. Other complementary works study how information about an input sample propagates through the network (Shwartz-Ziv & Tishby, 2017; Achille & Soatto, 2018; Saxe et al., 2019) or total amount of information (complexity) of a classification dataset (Lorena et al., 2019), rather than how much information the sample itself contains.

In terms of applications, our work is related to works that estimate influence of an example (Koh & Liang, 2017; Toneva et al., 2019; Katharopoulos & Fleuret, 2018; Ghorbani & Zou, 2019; Yoon et al., 2019). This can be done by estimating the change in weights if a sample is removed from the training set, which is addressed by several works (Koh & Liang, 2017; Golatkar et al., 2020; Wu et al., 2020). Influence functions (Cook, 1977; Koh & Liang, 2017) model removal of a sample as reducing its weight infinitesimally in the loss function, and show an efficient first-order approximation of its effect on other measures (such as test time predictions). We found influence functions to be prohibitively slow for the networks and data regimes we consider. Basu et al. (2020) found that influence functions are not accurate for large DNNs. Additionally, influence functions assume that the training has converged, which is not usually the case in practice. We instead use linearization of neural networks to estimate the effect of removing an example efficiently. We find that this approximation is accurate in realistic settings, and that the computational cost scales better with network size, making it applicable to very large neural networks.

Our work is orthogonal to that of feature selection: while we aim to evaluate the informativeness for the final weights of a subset of training samples, feature selection aims to quantify the informativeness for the task variable of a subset of features. However, they share some high-level similarities. In particular, Kohavi et al. (1997) propose the notion of strongly-relevant feature as one that changes the discriminative distribution when it is excluded. This notion is similar to the notion of unique sample information in eq. (1).

## 3 UNIQUE INFORMATION OF A SAMPLE IN THE WEIGHTS

Consider a (possibly stochastic) training algorithm $A$ that, given a training dataset $S$, returns the weights $W$ of a classifier $f_w$. From an information-theoretic point of view, the amount of unique information a sample $z_i = (x_i, y_i)$ provides about the weights is given by the conditional point-wise mutual information:

$$I(W; Z_i = z_i \mid \mathbf{S}_{-i} = S_{-i}) = \mathrm{KL}(p_A(w \mid \mathbf{S} = S) \,\|\, r(w \mid \mathbf{S}_{-i} = S_{-i})), \qquad (1)$$

where $\mathbf{S}$ denotes the random variable whose sample is the particular dataset $S$, and $r(w \mid \mathbf{S}_{-i} = S_{-i}) = \int p_A(w \mid \mathbf{S} = S_{-i}, z_i') dP(z_i') = \mathbb{E}_{z_i' \sim p(z)}[p_A(w \mid \mathbf{S} = S_{-i}, z_i')]$ denotes the marginal distribution of the weights over all possible sampling of $Z_i$.[1] Computing the distribution $r(w \mid S_{-i})$ is challenging because of the high-dimensionality and the cost of training algorithm $A$ for multiple samples. One can address this problem by using the following upper bound (Lemma B.1):

$$\mathrm{KL}(p_A(w \mid S) \,\|\, r(w \mid S_{-i})) = \mathrm{KL}(p_A(w \mid S) \,\|\, q(w \mid S_{-i})) - \mathrm{KL}(r(w \mid S_{-i}) \,\|\, q(w \mid S_{-i}))$$
$$\leq \mathrm{KL}(p_A(w \mid S) \,\|\, q(w \mid S_{-i})), \qquad (2)$$

---

[1]Hereafter, to avoid notational clutter, we will shorten "$\mathbf{S}_{-i} = S_{-i}$" to $S_{-i}$ and "$Z_i = z_i$" to just $z_i$ in all conditionals and information-theoretic functionals.

which is valid for any distribution $q(w \mid S_{-i})$. Choosing $q(w \mid S_{-i}) = p_A(w \mid S_{-i})$, the distribution of the weights after training on $S_{-i}$, gives a reasonable upper bound (see Sec. A.1 for details):

$$I(W; z_i \mid S_{-i}) \leq \mathrm{KL}(p_A(w \mid S) \,\|\, p_A(w \mid S_{-i})). \tag{3}$$

We call $\mathrm{SI}(z_i, A) \triangleq \mathrm{KL}(p_A(w \mid S) \,\|\, p_A(w \mid S_{-i}))$ the *sample information* of $z_i$ w.r.t. algorithm $A$.

**Smoothed Sample Information.** The formulation above is valid in theory but, in practice, even SGD is used in a deterministic fashion by fixing the random seed and, in the end, we obtain just one set of weights rather than a distribution of them. Under these circumstances, all the above KL divergences are degenerate, as they evaluate to infinity. It is common to address the problem by assuming that $A$ is a continuous stochastic optimization algorithm, such as stochastic gradient Langevin dynamics (SGLD) or a continuous approximation of SGD which adds Gaussian noise to the gradients. However, this creates a disconnect with the practice, where such approaches do not perform at the state-of-the-art. Our definitions below aim to overcome this disconnect.

**Definition 3.1** (Smooth sample information). *Let $A$ be a possibly stochastic algorithm. Following eq. (3), we define the* smooth sample information *with smoothing $\Sigma$:*

$$\boxed{\mathrm{SI}_\Sigma(z_i, A) = \mathrm{KL}(p_{A_\Sigma}(w \mid S) \,\|\, p_{A_\Sigma}(w \mid S_{-i})).} \tag{4}$$

*where we define smoothed weights $A_\Sigma(S) \triangleq A(S) + \xi$, with $\xi \sim \mathcal{N}(0, \Sigma)$.*

Note that if the algorithm $A$ is continuous, we can pick $\Sigma \to 0$, which will make $\mathrm{SI}_\Sigma(z_i, A) \to \mathrm{SI}(z_i, A)$. The following proposition shows how to compute the value of $\mathrm{SI}_\Sigma$ in practice.

**Proposition 3.2.** *Let $A$ be a deterministic training algorithm. Then, we have:*

$$\mathrm{SI}_\Sigma(z_i, A) = \frac{1}{2}(w - w_{-i})^T \Sigma^{-1}(w - w_{-i}), \tag{5}$$

*where $w = A(S)$ and $w_{-i} = A(S_{-i})$ are the weights obtained by training respectively with and without the training sample $z_i$. That is, the value of $\mathrm{SI}_\Sigma(z_i)$ depends on the distance between the solutions obtained training with and without the sample $z_i$, rescaled by $\Sigma$.*

The smoothing of the weights by a matrix $\Sigma$ can be seen as a form of soft-discretization. Rather than simply using an isotropic discretization $\Sigma = \sigma^2 I$ – since different filters have different norms and/or importance for the final output of the network – it makes sense to discretize them differently. In Sections 4 and 5 we show two canonical choices for $\Sigma$. One is the inverse of the Fisher information matrix, which discounts weights not used for classification, and the other is the covariance of the steady-state distribution of SGD, which respects the level of SGD noise and flatness of the loss.

## 4 UNIQUE INFORMATION IN THE PREDICTIONS

$\mathrm{SI}_\Sigma(z_i, A)$ measures how much information an example $z_i$ provides to the weights. Alternatively, instead of working in weight-space, we can approach the problem in function-space, and measure the informativeness of a training example for the network outputs or activations. The unique information that $z_i$ provides to the predictions on a test example $x$ is:

$$I(z_i; \widehat{y} \mid x, S_{-i}) = \mathbb{E}_S \,\mathrm{KL}(q(\widehat{y} \mid x, S) \,\|\, r(\widehat{y} \mid x, S_{-i})),$$

where $x \sim p(x)$ is a previously unseen test sample, $\widehat{y} \sim q(\cdot \mid x, S)$ is the network output on input $x$ after training on $S$, and $r(\widehat{y} \mid x, S_{-i}) = \int q(\widehat{y} \mid x, S_{-i}, z_i') dP(z_i') = \mathbb{E}_{z_i'} q(\widehat{y} \mid x, S_{-i}, z_i')$. Following the reasoning in the previous section, we arrive at

$$I(z_i; \widehat{y} \mid x, S_{-i}) \leq \mathrm{KL}(q(\widehat{y} \mid x, S) \,\|\, q(\widehat{y} \mid x, S_{-i})). \tag{6}$$

Again, when training with a discrete algorithm and/or when the output of the network is deterministic, the above quantity may be infinite. Similar to smooth sample information, we define:

**Definition 4.1** (Smooth functional sample information). *Let $A$ be a possibly stochastic training algorithm and let $\widehat{y}$ be the prediction on a test example $x$ after training on $S$. We define the* smooth functional sample information *(F-SI) as:*

$$\boxed{\text{F-SI}_\sigma(z_i, A) = \mathbb{E}_{(x,y)}[\mathrm{KL}(q_\sigma(\widehat{y}_\sigma \mid x, S) \,\|\, q_\sigma(\widehat{y}_\sigma \mid x, S_{-i}))],} \tag{7}$$

*where $\widehat{y}_\sigma = \widehat{y}(x, S) + n$, with $n \sim \mathcal{N}(0, \sigma^2 I)$ and $q_\sigma(\widehat{y}_\sigma \mid x, S)$ being the distribution of $\widehat{y}_\sigma$.*

We now describe a first-order approximation of the value of F-SI$_\Sigma$ for deterministic algorithms.

**Proposition 4.2.** *Let $A$ be a deterministic algorithm, $w = A(S)$ and $w_{-i} = A(S_{-i})$ be the weights obtained training respectively with and without sample $z_i$. Then,*

$$\text{F-SI}_\sigma(z_i, A) = \frac{1}{2\sigma^2} \mathbb{E}_{x \sim p(x)} \| f_w(x) - f_{w_{-i}}(x) \|_2^2 \tag{8}$$

$$\approx \frac{1}{2\sigma^2} (w - w_{-i})^T F(w)(w - w_{-i}), \tag{9}$$

*with $F(w) = \mathbb{E}_x \left[ \nabla_w f_w(x) \nabla_w f_w(x)^T \right]$ being the Fisher information matrix of $q_{\sigma=1}(\widehat{y} \mid x, S)$.*

By comparing eq. (5) and eq. (9), we see that the functional sample information is approximated by using the inverse of the Fisher information matrix to smooth the weight space. However, this smoothing is not isotropic as it depends on the point $w$.

## 5 EXACT SOLUTION FOR LINEARIZED NETWORKS

In this section, we derive a close-form expression for SI$_\Sigma$ and F-SI$_\Sigma$ using a linear approximation of the network around the initial weights. We show that this approximation can be computed efficiently and, as we validate empirically in Sec. 6, correlates well with the actual informativeness values. We also show that the covariance matrix of SGD's steady-state distribution is a canonical choice for the smoothing matrix $\Sigma$ of SI$_\Sigma$.

**Linearized Network.** Linearized neural networks are a class of neural networks obtained by taking the first-order Taylor expansion of a DNN around the initial weights (Lee et al., 2019):

$$f_w^{\text{lin}}(x) \triangleq f_{w_0}(x) + \nabla_w f_w(x)^T |_{w=w_0} (w - w_0).$$

These networks are linear with respect to their parameters $w$, but can be highly non-linear with respect to their input $x$. One of the advantages of linearized neural networks is that the dynamics of continuous-time or discrete-time gradient descent can be written analytically if the loss function is the mean squared error (MSE). In particular, for continuous-time gradient descent with constant learning rate $\eta > 0$, we have (Lee et al., 2019):

$$w_t = \nabla_w f_0(X) \Theta_0^{-1} \left( I - e^{-\eta \Theta_0 t} \right) (f_0(X) - Y), \tag{10}$$

$$f_t^{\text{lin}}(x) = f_0(x) + \Theta_0(x, X) \Theta_0^{-1} \left( I - e^{-\eta \Theta_0 t} \right) (Y - f_0(X)), \tag{11}$$

where $\Theta_0 = \nabla_w f_0(X)^T \nabla_w f_0(X) \in \mathbb{R}^{nk \times nk}$ is the Neural Tangent Kernel (NTK) (Jacot et al., 2018; Lee et al., 2019) and $\Theta_0(x, X) = \nabla_w f_0(x)^T \nabla_w f_0(X)$. The expressions for networks trained with weight decay is essentially the same (see Sec. F). To keep the notation simple, we will use $f_w(x)$ to indicate $f_w^{\text{lin}}(x)$ from now on.

**Stochastic Gradient Descent.** As mentioned in Sec. 3, a popular alternative approach to make information quantities well-defined is to use continuous-time SGD, which is defined by (Li et al., 2017; Mandt et al., 2017):

$$dw_t = -\eta \nabla_w \mathcal{L}_w(w_t) dt + \eta \sqrt{\frac{1}{b} \Lambda(w_t)} dn(t), \tag{12}$$

where $\eta$ is the learning rate, $b$ is the batch size, $n(t)$ is a Brownian motion, and $\Lambda(w_t)$ is the co-variance matrix of the per-sample gradients (see Sec. C for details). Let $A_{\text{SGD}}$ be the algorithm that returns a random sample from the steady-state distribution of (12), and let $A_{\text{ERM}}$ be the deterministic algorithm that returns the global minimum $w^*$ of the loss $\mathcal{L}(w)$ (for a regularized linearized network $\mathcal{L}(w)$ is strictly convex). We now show that the non-smooth sample information $\text{SI}(z_i, A_{\text{SGD}})$ is the same as the smooth sample information using SGD's steady-state covariance as the smoothing matrix and $A_{\text{ERM}}$ as the training algorithm.

**Proposition 5.1.** *Let the loss function be regularized MSE, $w^*$ be the global minimum of it, and algorithms $A_{SGD}$ and $A_{ERM}$ be defined as above. Assuming $\Lambda(w)$ is approximately constant around $w^*$ and SGD's steady-state covariance remains constant after removing an example, we have*

$$\text{SI}(z_i, A_{SGD}) = \text{SI}_\Sigma(z_i, A_{ERM}) = \frac{1}{2} (w^* - w_{-i}^*)^T \Sigma^{-1} (w^* - w_{-i}^*), \tag{13}$$

*where $\Sigma$ is the solution of*

$$H\Sigma + \Sigma H^T = \frac{\eta}{b}\Lambda(w^*), \tag{14}$$

*with $H = (\nabla_w f_0(X)\nabla_w f_0(X)^T + \lambda I)$ being the Hessian of the loss function.*

This proposition motivates the use of SGD's steady-state covariance as a smoothing matrix. From equations (13) and (14) we see that SGD's steady-state covariance is proportional to the flatness of the loss at the minimum, the learning rate, and to SGD's noise, while inversely proportional to the batch size. When $H$ is positive definite, as in our case when using weight decay, the continuous Lyapunov equation (14) has a unique solution, which can be found in $O(d^3)$ time using the Bartels-Stewart algorithm (Bartels & Stewart, 1972). One particular case when the solution can be found analytically is when $\Lambda(w^*)$ and $H$ commute, in which case $\Sigma = \frac{\eta}{2b}\Lambda H^{-1}$. For example, this is the case for Langevin dynamics, for which $\Lambda(w) = \sigma^2 I$ in equation (12). In this case, we have

$$\mathrm{SI}(z_i, A_{\mathrm{SGD}}) = \mathrm{SI}_\Sigma(z_i, A_{\mathrm{ERM}}) = \frac{b}{\eta\sigma^2}(w^* - w^*_{-i})^T H(w^* - w^*_{-i}), \tag{15}$$

which was already suggested by Cook (1977) as a way to measure the importance of a datum in linear regression.

**Functional Sample Information.** The definition in Section 4 simplifies for linearized neural networks: The step from eq. (8) to eq. (9) becomes exact, and the Fisher information matrix becomes independent of $w$ and equal to $F = \mathbb{E}_{x \sim p(x)}\left[\nabla_w f_0(x)\nabla_w f_0(x)^T\right]$. This shows that functional sample information can be seen as weight sample information with discretization $\Sigma$ equal to $F^{-1}$. The functional sample information depends on the training data distribution, which is usually unknown. We can estimate it using a validation set:

$$\mathrm{F\text{-}SI}_\sigma(z_i, A) \approx \frac{1}{2\sigma^2 n_{\mathrm{val}}}\sum_{j=1}^{n_{\mathrm{val}}}\left\| f_w(x_j^{\mathrm{val}}) - f_{w_{-i}}(x_j^{\mathrm{val}})\right\| \tag{16}$$

$$= \frac{1}{2\sigma^2 n_{\mathrm{val}}}(w - w_{-i})^T(H_{\mathrm{val}} - \lambda I)(w - w_{-i}). \tag{17}$$

It is instructive to compare the sample weight information of (15) and functional sample information of (17). Besides the constants, the former uses the Hessian of the training loss, while the latter uses the Hessian of the validation loss (without the $\ell_2$ regularization term). One advantage of the latter is computational cost: As demonstrated in the next section, we can use equation (16) to compute the prediction information, entirely in the function space, without any costly operation on weights. For this reason, we focus on the linearized F-SI approximation in our experiments. Since $\sigma^{-2}$ is just a multiplicative factor in (17) we set $\sigma = 1$. We also focus on the case where the training algorithm $A$ is discrete gradient descent running for $t$ epochs (equations 10 and 11).

**Efficient Implementation.** To compute the proposed sample information measures for linearized neural networks, we need to compute the change in weights $w - w_{-i}$ (or change in predictions $f_w(x) - f_{w_i}(x)$) after excluding an example from the training set. This can be done without retraining using the analytical expressions of weight and prediction dynamics of linearized neural networks eq. (10) and eq. (11), which also work when the algorithm has not yet converged ($t < \infty$). We now describe a series of measures to make the problem tractable. First, to compute the NTK matrix we would need to store the Jacobian $\nabla f_0(x_i)$ of all training points and compute $\nabla_w f_0(X)^T\nabla_w f_0(X)$. This is prohibitively slow and memory consuming for large DNNs. Instead, similarly to Zancato et al. (2020), we use low-dimensional random projections of per-example Jacobians to obtain provably good approximations of dot products (Achlioptas, 2003; Li et al., 2006). We found that just taking 2000 random weights coordinates per layer provides a good enough approximation of the NTK matrix. Importantly, we consider each layer separately, as different layers may have different gradient magnitudes. With this method, computing the NTK matrix takes $O(nkd + n^2k^2 d_0)$ time, where $d_0 \approx 10^4$ is the number of sub-sampled weight indices ($d_0 \ll d$). We also need to recompute $\Theta_0^{-1}$ after removing an example from the training set. This can be done in quadratic time by using rank-one updates of the inverse (see Sec. E). Finally, when $t \neq \infty$ we need to recompute $e^{-\eta\Theta_0 t}$ after removing an example. This can be done in $O(n^2k^2)$ time by downdating the eigen-decomposition of $\Theta_0$ (Gu & Eisenstat, 1995). Overall, the complexity of computing $w - w_i$ for all training examples is $O(n^2k^2 d_0 + n(n^2k^2 + C))$, $C$ is the complexity of a single pass over the

| | Reg. | Method | MNIST MLP | MNIST CNN | | Cats and Dogs | |
|---|---|---|---|---|---|---|---|
| | | | scratch | scratch | pretrained | pr. ResNet-18 | pr. ResNet-50 |
| weights | $\lambda = 0$ | Linearization | **0.987** | 0.193 | **0.870** | **0.895** | **0.968** |
| | | Infl. functions | 0.935 | **0.319** | 0.736 | 0.675 | 0.897 |
| | $\lambda = 10^3$ | Linearization | **0.977** | -0.012 | 0.964 | **0.940** | 0.816 |
| | | Infl. functions | **0.978** | **0.069** | **0.979** | 0.858 | **0.912** |
| predictions | $\lambda = 0$ | Linearization | **0.993** | 0.033 | **0.875** | **0.877** | **0.895** |
| | | Infl. functions | 0.920 | **0.647** | 0.770 | 0.530 | 0.715 |
| | $\lambda = 10^3$ | Linearization | **0.993** | 0.070 | **0.974** | **0.931** | **0.519** |
| | | Infl. functions | **0.990** | **0.407** | 0.954 | 0.753 | 0.506 |

Table 1: Pearson correlations of weight change $\|w - w_{-i}\|_2^2$ and validation prediction change $\|f_w(X_{\text{val}}) - f_{w_{-i}}(X_{\text{val}})\|_2^2$ norms computed with influence functions and linearized neural networks with their corresponding measures computed for standard neural networks with retraining.

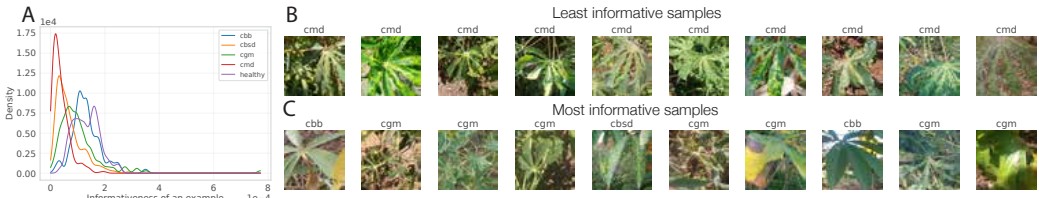

Figure 1: Functional sample information of samples in the iCassava classification task with 1000 samples, where the network is a pretrained ResNet-18. **A**: histogram of sample informations, **B**: 10 least informative samples, **C**: 10 most informative samples.

training dataset. The complexity of computing functional sample information for $m$ test samples is $O(C + nmk^2d_0 + n(mnk^2 + n^2k^2))$. This depends on the network size lightly, only through $C$.

## 6 EXPERIMENTS

In this section, we test the validity of linearized network approximation in terms of estimating the effects of removing an example and show several applications of the proposed information measures. Additional results and details are provided in the supplementary Sec. A.

**Accuracy of the linearized network approximation.** We measure $\|w - w_{-i}\|_2^2$ and $\|f_w(X_{\text{val}}) - f_{w_{-i}}(X_{\text{val}})\|_2^2$ for each sample $z_i$ by training with and without that example. Then, instead of retraining, we use the efficient linearized approximation in Sec. 6 to estimate the same quantities and measure their correlation with the ground-truth values (Table 1). For comparison, we also estimate these quantities using influence functions (Koh & Liang, 2017). We consider two classification tasks: (a) a toy MNIST 4 vs 9 classification task and (b) Kaggle Dogs vs. Cats classification task (Kaggle, 2013), both with 1000 examples. For MNIST we consider a fully connected network with a single hidden layer of 1024 ReLU units (MLP) and a small 4-layer convolutional network (CNN), either trained from scratch or pretrained on EMNIST letters (Cohen et al., 2017). For cats vs dogs classification, we consider ResNet-18 and ResNet-50 networks (He et al., 2016) pretrained on ImageNet. In both tasks, we train both with and without weight decay ($\ell_2$ regularization). The results in Table 1 shows that linearized approximation correlates well with ground-truth when the network is wide enough (MLP) and/or pretraining is used (CNN with pretraining and pretrained ResNets). This is expected, as wider networks can be approximated with linearized ones better (Lee et al., 2019), and pretraining decreases the distance from initialization, making the Taylor approximation more accurate. Adding regularization also keeps the solution close to initialization, and generally increases the accuracy of the approximation. Furthermore, in most cases linearization gives better results compared to influence functions, while also being around 30 times faster in our settings.

**Which examples are informative?** Fig. 1, and Fig. 4 of the supplementary, plot the top 10 least and most important samples in iCassava plant disease classification (Mwebaze et al., 2019), the MNIST 4 vs 9, and Kaggle cats vs dogs classification tasks. Especially in the case of the last two, we see that the least informative samples look typical and easy, while the most informative ones look more

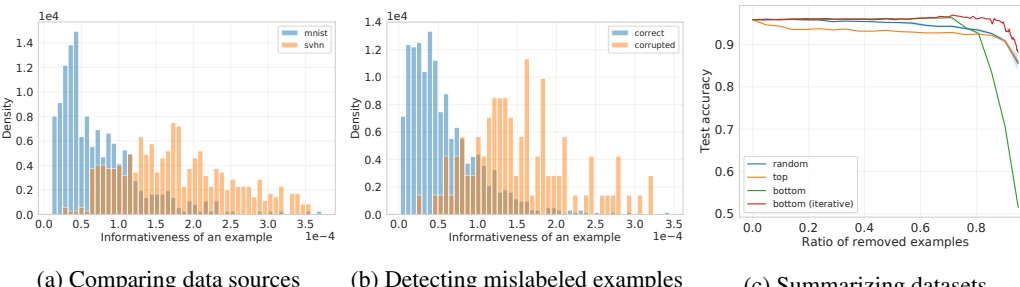

(a) Comparing data sources      (b) Detecting mislabeled examples      (c) Summarizing datasets

Figure 2: Applications of functional sample information. (a) Different sources of data for the same task (digit classification) can have a vastly different amount of information. (b) As expected, samples with the wrong labels carry more unique information. (c) Test accuracy as a function of the ratio of removed training examples using different strategies.

challenging and atypical. In the case of iCassava, the informative samples are more zoomed on features that are important for classification (e.g., the plant disease spots). We observe that most samples have small unique information, possibly because they are easier or because the dataset may have many similar-looking examples. While in the case of MNIST 4 vs 9 and Cats vs. Dogs, the two classes have on average similar information scores, in Fig. 1a we see that in iCassava examples from rare classes (such as 'healthy' and 'cbb') are on average more informative.

**Which data source is more informative?** In this experiment, we train for 10-way digit classification where both the training and validation sets consist of 1000 samples, 500 from MNIST and 500 from SVHN. We compute functional sample information for a pretrained ResNet-18 network. The results presented in Fig. 2a tell that SVHN examples are much more informative than MNIST examples. The same result holds if we change the network to be a pretrained DenseNet-121 (see Fig. 7a). This is intuitive, as SVHN examples have more variety. One can go further and use our information measures for estimating informativeness of examples of dataset $A$ for training a classifier for task $B$, similar to Torralba & Efros (2011).

**Detecting mislabeled examples.** We expect a mislabeled example to carry more unique information, since the network needs to memorize unique features of that particular example to classify it. To test this, we add 10% uniform label noise to MNIST 4 vs 9, Kaggle cats vs dogs, and iCassava classification tasks (all with 1000 examples in total), while keeping the validation sets clean. Fig. 2b plots the histogram of functional sample information for both correct and mislabeled examples in the case of iCassava classification task with a pretrained ResNet-18, while Fig. 8a and 8b plot that for MNIST 4 vs 9 and Kaggle cats vs dogs tasks, respectively. The results indicate that mislabeled examples are much more informative on average. This suggests that our information measures can be used to detect outliers or corrupted examples.

**Data summarization.** We subsample the training dataset by removing a fraction of the least informative examples and measure the test performance of the resulting model. We expect that removing the least informative training samples should not affect the performance of the model. Note however that, since we are considering the *unique* information, removing one sample can increase the informativeness of another. For this reason, we consider two strategies: In one we compute the informativeness scores once, and remove a given percentage of the least informative samples. In the other we remove 5% of the least informative samples, recompute the scores, and iterate until we remove the desired number of samples. For comparison, we also consider removing the most informative examples ("top" baseline) and randomly selected examples ("random" baseline). The results on MNIST 4 vs 9 classification task with the one-hidden-layer network described earlier, are shown in Fig. 2c. Indeed, removing the least informative training samples has little effect on the test error, while removing the top examples has the most impact. Also, recomputing the information scores after each removal steps ("bottom iterative") greatly improves the performance when many samples are removed, confirming that SI and F-SI are good practical measures of unique information in a sample, but also that the total information in a large group is not simply the sum of the unique information of its samples. Interestingly, removing more than 80% of the least informative examples degrades the performance more than removing the same number of the most informative examples. In the former case, we are left with a small number of most informative examples, some

of which are outliers, while in the latter case we are left with the same number of least informative examples, most of which are typical. Consequently, the performance is better in the latter case.

**Detecting under-sampled sub-classes.** Using CIFAR-10 images, we create a dataset of "Pets vs Deer": Pets has 4200 samples and deer 4000. The class "pets" consists of two unlabeled sub-classes, cats (200) and dogs (4000). Since there are relatively few cat images, we expect each to carry more unique information. This is confirmed when we compute F-SI for a pretrained ResNet-18 (see Fig. 9 of the supplementary), suggesting that the F-SI can help to detect when an unlabeled sub-class of a larger class is under-sampled.

## 7    DISCUSSION AND FUTURE WORK

The smooth (functional) sample information depends not only on the example itself, but on the network architecture, initialization, and the training procedure (i.e. the training algorithm). This has to be the case, since an example can be informative with respect to one algorithm or architecture, but not informative for another one. Similarly, some examples may be more informative at the beginning of the training (e.g., simpler examples) rather than at the end. Nevertheless, we found out that F-SI still captures something inherent in the example. In particular, as shown Sec. A.4 of the supplementary, F-SI scores computed with respect to different architectures are significantly correlated to each other (e.g. around 45% correlation in case of ResNet-50 and DenseNet-121). Furthermore, F-SI scores computed for different initializations are significantly correlated (e.g. around 36%). Similarly, F-SI scores computed for different training lengths are strongly correlated (see Fig. 5b). This suggests that F-SI computed with respect to one network can reveal useful information for another one. Indeed, this is verified in Sec. A.4 of the supplementary, where we redo the data summarization experiment, but with a slight change that F-SI scores are computed for one network, but another network is trained to check the accuracy. As shown in the Fig. 7b the results look qualitatively similar to those of the original experiment.

The proposed sample information measure only the unique information provided by an example. For this reason, it is not surprising that typical examples are usually the least informative, while atypical and rare ones are more informative. This holds not only for visual data (as shown above), but also for textual data (see Sec. A.8 of the supplementary). While the typical examples are usually less informative according to the proposed measures, they still provide information about the decision functions, which is evident in the data summarization experiment – removing lots of typical examples was worse than removing the same number of random examples. Generalizing sample information to capture this kind of contributions is an interesting direction for future work. Similar to Data Shapley (Ghorbani & Zou, 2019)), one can look at the average unique information an example provides when considering along with a random subset of data. One can also consider common information, high-order information, synergistic information, and other notions of information *between* samples. The relation among these quantities is complex in general, even for 3 variables (Williams & Beer, 2010) and is an open challenge.

## 8    CONCLUSION

There are many notions of information that are relevant to understanding the inner workings of neural networks. Recent efforts have focused on defining information in the weights or activations that do not degenerate for deterministic training. We look at the information in the training data, which ultimately affects both the weights and the activations. In particular, we focus on the most elementary case, which is the unique information contained in a sample, because it can be the foundation for understanding more complex notions. However, our approach can be readily generalized to unique information of a group of samples. Unlike most previously introduced information measures, ours is tractable even for real datasets used to train standard network architectures, and does not require restriction to limiting cases. In particular, we can approximate our quantities without requiring the limit of small learning rate (continuous training time), or the limit of infinite network width.

ACKNOWLEDGMENTS

We thank the anonymous reviewers whose comments/suggestions helped improve and clarify this manuscript.

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

## A  ADDITIONAL RESULTS AND DETAILS

In this section we present additional results and details that were not included in the main paper due to the space constraint.

### A.1  APPROXIMATING UNIQUE INFORMATION WITH LEAVE-ONE-OUT KL DIVERGENCE

In the main text we discussed that $I(W; z_i \mid S_{-i})$ can be upper bounded with $\mathrm{KL}(p_A(w \mid S) \,\|\, p_A(w \mid S_{-i}))$. In this subsection we evaluate these quantities on a toy 2D dataset. The dataset has two classes, each with 40 examples, generated from a Gaussian distribution (see Fig. 3a). We consider training a linear regression on this dataset using stochastic gradient descent for 200 epochs, with batch size equal to 5 and 0.1 learning rate. Fig. 3b plots the distribution $p_A(w \mid S)$, Fig. 3c the distribution $p_A(w \mid S_{-i})$, while Fig. 3d plots the distribution $\mathbb{E}_{z_i'} p_A(w \mid S_{-i}, z_i')$. On this example we get $I(W; z_i \mid S_{-i}) \approx 1.3$ and $\mathrm{KL}(p_A(w \mid S) \,\|\, p_A(w \mid S_{-i})) \approx 3.0$.

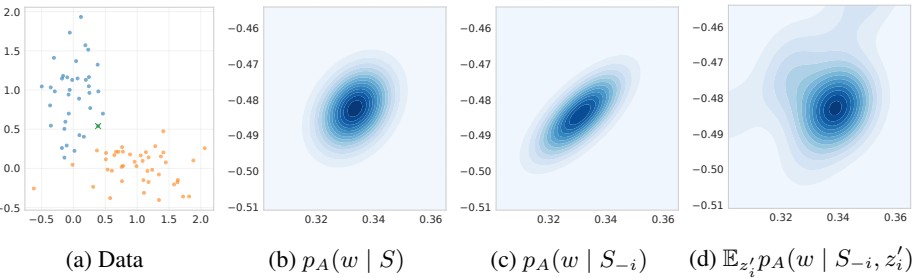

     (a) Data         (b) $p_A(w \mid S)$      (c) $p_A(w \mid S_{-i})$    (d) $\mathbb{E}_{z_i'} p_A(w \mid S_{-i}, z_i')$

Figure 3: A toy dataset and key distributions involved in upper bounding the unique sample information with leave-one-out KL divergence.

## A.2 WHICH EXAMPLES ARE INFORMATIVE?

In this subsection we present the most and least informative examples for two more classification tasks: MNIST 4 vs 9 classification with MLP and Kaggle cat vs dog classification with a pretrained ResNet-18 (Fig. 4). The results indicate that most informative examples are often the challenging and atypical ones, while the least informative ones are easy and typical ones. For the MLP network on MNIST 4 vs 9 we set $t = 2000$ and $\eta = 0.001$; for the ResNet-18 on cats vs dog classification $t = 1000$ and $\eta = 0.001$; and for the ResNet-18 on iCassava dataset $t = 5000$ and $\eta = 0.0003$.

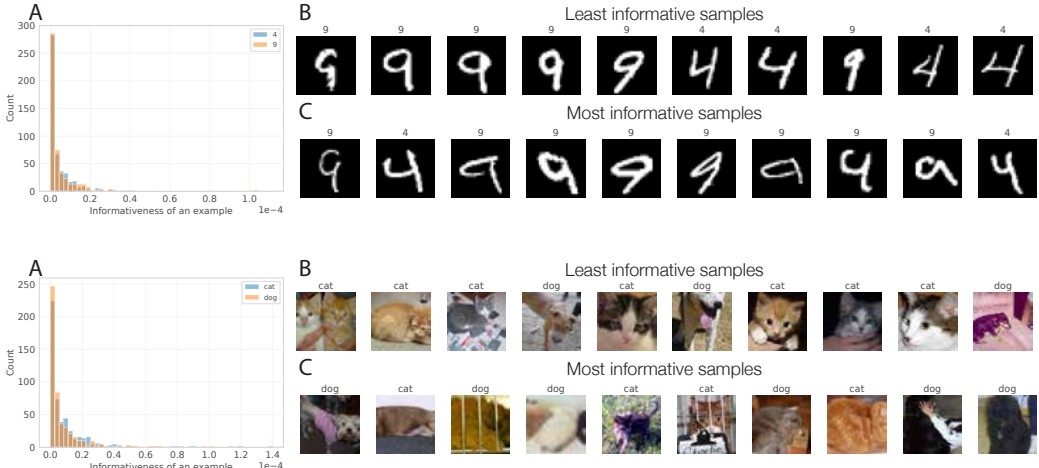

Figure 4: Functional sample information of samples in MNIST 4 vs 9 classification task (top) and Dogs vs. Cats (bottom), with **A**: histogram of sample informations, **B**: 10 least informative samples, **C**: 10 most informative samples.

## A.3 ACCURACY OF THE LINEARIZED NETWORK APPROXIMATION

In this subsection we present additional details of experiments presented in Table 1. In all datasets used the validation set also has 1000 samples. The fully connected network, MLP, consists of a single layer of 1024 ReLU units. The architecture of the CNN used for the experiments in Table 1 is as follow.

| Layer | Layer parameters |
|---|---|
| Conv. 1 | 32 filters, 4x4 kernel, stride = 2, padding = 1, ReLU activation |
| Conv. 2 | 32 filters, 4x4 kernel, stride = 2, padding = 1, ReLU activation |
| Conv. 3 | 64 filters, 3x3 kernel, stride = 1, padding = 0, ReLU activation |
| Conv. 4 | 256 filters, 3x3 kernel, stride = 1, padding = 0, ReLU activation |
| Fully connected | 256 ReLU neurons |
| Fully connected | 1 linear unit |

In all our experiments, when using pretrained ResNets, we disable the exponential averaging of batch statistics in batch norm layers. The exact details of running influence functions and linearized neural network predictions are presented in Table 2.

## A.4 HOW MUCH DOES SAMPLE INFORMATION DEPEND ON ALGORITHM?

The proposed information measures depend on the training algorithm, which includes the architecture, seed, initialization, and training length. This is unavoidable as one example can be more informative for one algorithm and less informative for another. Nevertheless, in this subsection, we test how much does informativeness depend on the network, initialization, and training time. We consider the Kaggle cats vs dogs classification task with 1000 training examples. First, fixing the training time $t = 1000$, we consider four pretrained architectures: ResNet-18, ResNet-34, ResNet-50, and DenseNet-121. The correlations between F-SI scores computed for the four architectures

| Experiment | Method | Details |
|---|---|---|
| MNIST MLP (scratch) | Brute force
Infl. functions
Linearization | 2000 epochs, learning rate = 0.001, batch size = 1000
LiSSA algorithm, 1000 recursion steps, scale = 1000
$t = 2000$, learning rate = 0.001 |
| MNIST CNN (scratch) | Brute force
Infl. functions
Linearization | 1000 epochs, learning rate = 0.01, batch size = 1000
LiSSA algorithm, 1000 recursion steps, scale = 1000
$t = 1000$, learning rate = 0.01 |
| MNIST CNN (pretrained) | Brute force
Infl. functions
Linearization | 1000 epochs, learning rate = 0.002, batch size = 1000
LiSSA algorithm, 1000 recursion steps, scale = 1000
$t = 1000$, learning rate = 0.002 |
| Cats and dogs | Brute force
Infl. functions
Linearization | 500 epochs, learning rate = 0.001, batch size = 500
LiSSA algorithm, 50 recursion steps, scale = 1000
$t = 1000$, learning rate = 0.001 |

Table 2: Details of experiments presented in Table 1. For influence functions, we add a dumping term with magnitude 0.01 whenever $\ell_2$ regularization is not used (i.e. $\lambda = 0$).

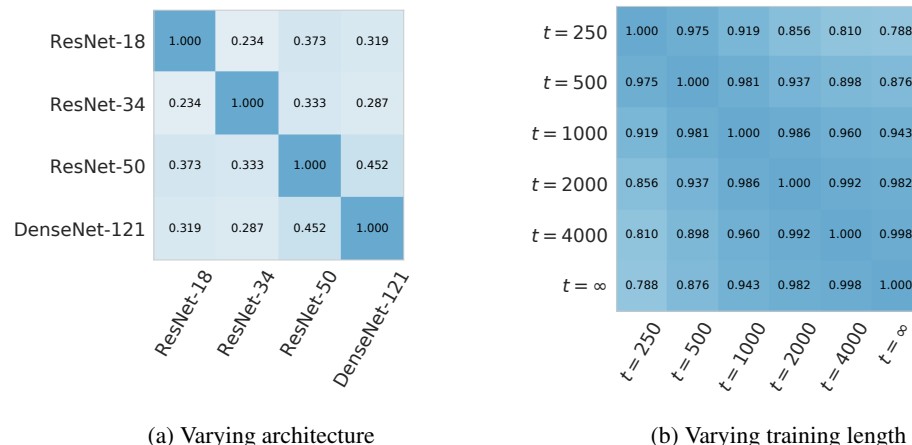

(a) Varying architecture  (b) Varying training length

Figure 5: Correlations between functional sample information scores computed for different architectures and training lengths. **On the left**: correlations between F-SI scores of the 4 pretrained networks, all computed with setting $t = 1000$ and $\eta = 0.001$. **On the right:** correlations between F-SI scores computed for pretrained ResNet-18s, with learning rate $\eta = 0.001$, but varying training lengths $t$. All reported correlations are averages over 10 different runs. The training dataset consists of 1000 examples from the Kaggle cats vs dogs classification task.

are presented in Fig. 5a. We see that F-SI scores computed for two completely different architectures, such as ResNet-50 and DenseNet-121 have significant correlation, around 45%. Furthermore, there is a significant overlap in top 10 most informative examples for these networks (see Fig. 6). Next, fixing the network to be a pretrained ResNet-18 and fixing the training length $t = 1000$, we consider changing initialization of the classification head (which is not pretrained). In this case the correlation between F-SI scores is $0.364 \pm 0.066$. Finally, fixing the network to be a ResNet-18 and fixing the initialization of the classification head, we consider changing the number of iterations in the training. We find strong correlations between F-SI scores of different training lengths (see Fig. 5b).

**MNIST vs SVHN experiment for DenseNet-121.** In this paragraph we redo the MNIST vs SVHN experiment (Fig 2a) but for a different network, a pretrained DenseNet-121, to test the dependence of the results on the architecture choice. The results are presented in Fig. 7a and are qualitatively identical to the results derived with a pretrained ResNet18 (Fig. 2a).

**Data summarization with a change of architecture.** To test how much sample information scores computed for one network are useful for another network, we reconsider the MNIST 4 vs 9 data summarization experiment (Fig. 2c). This time we compute F-SI scores for the original net-

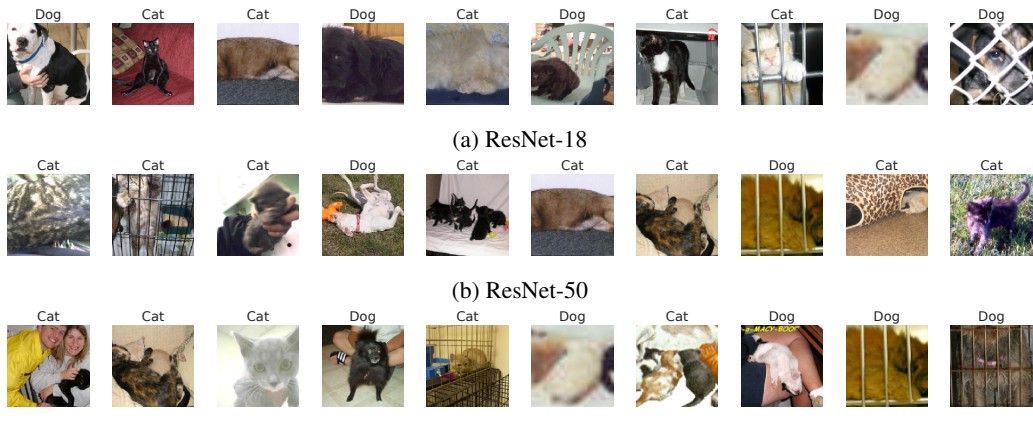

(a) ResNet-18

(b) ResNet-50

(c) DenseNet-121

Figure 6: Top 10 most informative examples from Kaggle cats vs dogs classification task for three pretrained networks: ResNet-18, ResNet-50, and DenseNet-121.

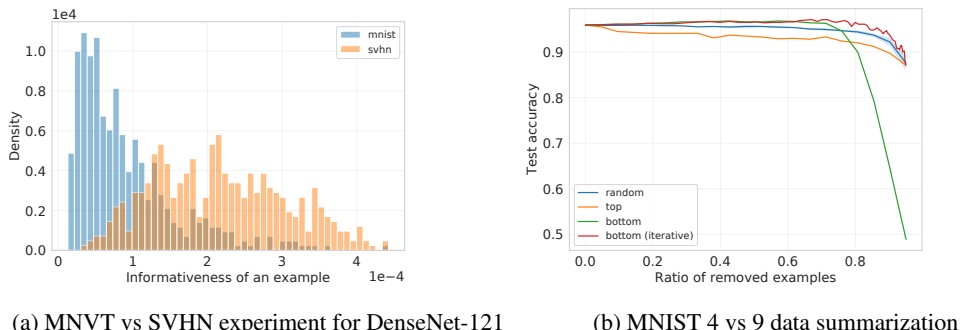

(a) MNVT vs SVHN experiment for DenseNet-121

(b) MNIST 4 vs 9 data summarization

Figure 7: Testing how much F-SI scores computed for different networks are qualitatively different. **On the left:** the MNIST vs SVHN experiment with a pretrained DesneNet-121 instead of a pretrained ResNet-18. **On the right:** Data summarization for the MNIST 4 vs 9 classification task, where the F-SI scores are computed for a one-hidden-layer network, but a two-hidden-layer network is trained to produce the test accuracies.

work with one hidden layer, but train a two-hidden-layer neural network (both layers having 1024 ReLU units). The data summarization results presented in Fig. 7b are qualitatively and quantitively almost identical to the original results presented in the main text. This confirms that F-Si scores computed for one network can be useful for another network.

### A.5 DETECTING INCORRECT EXAMPLES

In this subsection we present results on detecting mislabeled examples for two more classification tasks: MNIST 4 vs 9 classification with an MLP (Fig. 8a) and Kaggle cat vs dog classification with a pretrained ResNet-18 (Fig. 8b). The results are qualitatively the same and assert that, indeed, mislabeled examples are more informative, on average. For the MLP network we set $t = 2000$ and $\eta = 0.001$; for the ResNet-18 on the cats vs dog classification task $t = 1000$ and $\eta = 0.001$; and for the ResNet-18 on the iCassava dataset (in the main text) we set $t = 10000$ and $\eta = 0.0001$.

### A.6 DETECTING UNDER-SAMPLED SUB-CLASSES

Using CIFAR-10 images, we create a dataset of "Pets vs Deer": Pets has 4200 samples and deer 4000. The class pets consists of two unlabeled sub-classes, cats (200) and dogs (4000). Since there are relatively few cat images, we expect each to carry more unique information. Indeed, Fig. 9

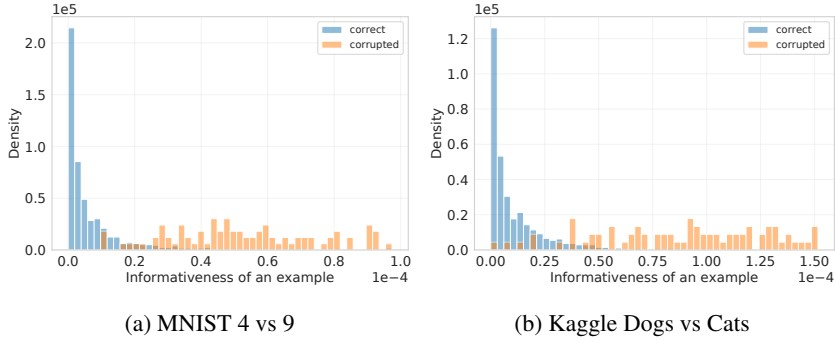

(a) MNIST 4 vs 9            (b) Kaggle Dogs vs Cats

Figure 8: Additional results comparing the information content of samples with correct and incorrect labels on MNIST 4 vs 9 and Dogs vs Cats.

shows that this is the case, suggesting that the F-SI can help detecting when an unlabeled sub-class of a larger class is under-sampled. In this experiment we set $t = 10000$ and $\eta = 0.0001$.

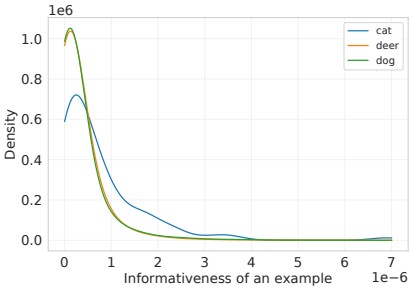

Figure 9: Histogram of the functional sample information of samples from the three subclasses of the Pets vs Deer. Since the sub-class "cat" is under-represented in the dataset, cat images tend to have on average more unique information than dog images, even if they belong to the same class.

## A.7 DETECTING ADVERSARIAL EXAMPLES

Szegedy et al. (2014) shows that imperceptible perturbation to an image can fool a neural network into predicting the wrong label (adversarial examples). Goodfellow et al. (2015) further shows that adding adversarial examples to the training dataset improves adversarial robustness and generalization (adversarial training), which suggests that adversarial examples may be informative for the training process. To test how informative adversarial examples are with respect to normal ones, we consider the Kaggle Cats vs Dogs classification task, consisting of 1000 examples. On this task we fine-tune a pretrained ResNet-18 and for 10% of examples create successful adversarial examples using the FGSM method of (Goodfellow et al., 2015) with $\epsilon = 0.01$. Then, for these 10% of examples, we replace the original images with the corresponding adversarial ones. With the resulting dataset, we consider a new pretrained ResNet-18 and compute F-SI for all training examples, setting $t = 1000$ and $\eta = 0.001$. The results reported in Fig. 10 confirm that adversarial examples are on average more informative than normal ones. Furthermore, the results indicate that one can use F-SI to successfully detect adversarial examples.

## A.8 INFORMATIVE EXAMPLES IN SENTIMENT ANALYSIS

So far we focused on image classification tasks. To test whether the proposed method works for other modalities, we consider a sentiment analysis task, consisting of 2000 samples from the IMDB movie reviews dataset (Maas et al., 2011). We convert words to 300-dimensional vectors using the pretrained GloVe 6B word vectors (Pennington et al., 2014). Then each review is mapped to a vector by averaging all of its word vectors. On the resulting vector we apply a fully connected neural

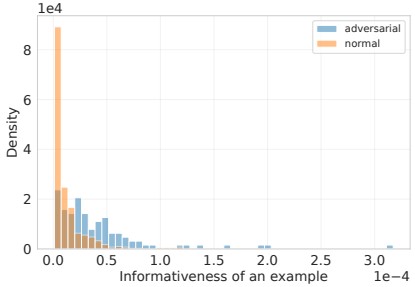

Figure 10: Histogram of the functional sample information of samples of the Kaggle cats vs dogs classification task, where 10% of examples are adversarial examples.

network with one hidden layer, consisting of 2000 ReLU units. We then compute F-SI scores for all training examples, setting $t = 10000$ and $\eta = 0.003$. Likewise to the case of images, we find that least informative examples are typical and easy (often containing many sentiment-specific words), while the most informative examples are more nuanced and hard. Additionally, we find that there is -16% correlation between the informativeness and length of review, possibly because averaging many word vectors makes long reviews similar to each other.

### A.9 REMAINING DETAILS

In the experiment of measuring which data source is more informative (Fig. 2a) we set $t = 20000$ and $\eta = 0.0001$. In the experiment of data summarization (Fig. 2c) we set $t = 2000$ and $\eta = 0.001$.

## B PROOFS

The proof of eq. (2) follows from the following more general lemma, applied to the conditional distribution $p_A(w|S_{-i}, z_i)$ and its marginal $r(x|S_{-i})$.

**Lemma B.1.** *Let $p(x|y)$ be a conditional probability density function, and let $p(x) = \int p(x|y)dP(y)$ denote its marginal distribution. Then, for any distribution $q(x)$ we have*

$$\mathrm{KL}(p(x|y) \,\|\, p(x)) = \mathrm{KL}(p(x|y) \,\|\, q(x)) - \mathrm{KL}(p(x) \,\|\, q(x))$$
$$\leq \mathrm{KL}(p(x|y) \,\|\, q(x)),$$

*where* $\mathrm{KL}(p(x|y) \,\|\, q(x)) \triangleq \int \int \log \frac{p(x|y)}{q(x)} p(x|y) dx dP(y)$ *denotes the conditional version of the KL divergence (Cover & Thomas, 2006, Section 2.5).*

*Proof.* The last inequality follows from the fact that the KL-divergence is always non-negative. We now prove the first equality:

$$\mathrm{KL}(p(x|y) \,\|\, q(x)) - \mathrm{KL}(p(x) \,\|\, q(x))$$
$$= \int \int \log \frac{p(x|y)}{q(x)} p(x|y) dx dP(y) - \int \log \frac{p(x)}{q(x)} p(x) dx$$
$$\overset{(a)}{=} \int \int \log \frac{p(x|y)}{q(x)} p(x|y) dx dP(y) - \int \log \frac{p(x)}{q(x)} \left( \int p(x|y) dP(y) \right) dx$$
$$\overset{(b)}{=} \int \int \log \frac{p(x|y)}{q(x)} p(x|y) dx dP(y) - \int \int \log \frac{p(x)}{q(x)} p(x|y) dx dP(y)$$
$$= \int \int \left[ \log \frac{p(x|y)}{q(x)} - \log \frac{p(x)}{q(x)} \right] p(x|y) dx dP(y)$$
$$= \int \int \log \frac{p(x|y)}{p(x)} p(x|y) dx dP(y)$$
$$= \mathrm{KL}(p(x|y) \,\|\, p(x)),$$

where in (a) we use that by definition $p(x) = \int p(x|y)dP(y)$ and in (b) we exchange the order of integration. □

Propositions 3.2 and 4.2 are trivial, since they just assert that KL divergence between two Gaussian distributions with equal covariance matrices $\Sigma$ and with means $\mu_1$ and $\mu_2$ is equal to $\frac{1}{2}(\mu_1 - \mu_2)^T\Sigma^{-1}(\mu_1 - \mu_2)$. We present the proof of Proposition 5.1 here.

**Proposition B.2** (Prop. 5.1 restated). *Let the loss function be regularized MSE, $w^*$ be the global minimum of it, and algorithms $A_{SGD}$ and $A_{ERM}$ be defined as in the main text. Assuming $\Lambda(w)$ is approximately constant around $w^*$ and SGD's steady-state covariance remains constant after removing an example, we have*

$$\text{SI}(z_i, A_{SGD}) = \text{SI}_\Sigma(z_i, A_{ERM}) = \frac{1}{2}(w^* - w^*_{-i})^T\Sigma^{-1}(w^* - w^*_{-i}), \tag{18}$$

*where $\Sigma$ is the solution of*

$$H\Sigma + \Sigma H^T = \frac{\eta}{b}\Lambda(w^*), \tag{19}$$

*with $H = (\nabla_w f_0(X)\nabla_w f_0(X)^T + \lambda I)$ being the Hessian of the loss function.*

*Proof.* Assuming $\Lambda(w)$ is approximately constant around $w^*$, the steady-state distributions of (12) is a Gaussian distribution with mean $w^*$ and covariance $\Sigma$ such that:

$$H\Sigma + \Sigma H^T = \frac{\eta}{b}\Lambda(w^*), \tag{20}$$

where $H = (\nabla_w f_0(X)\nabla_w f_0(X)^T + \lambda I)$ is the Hessian of the loss function (Mandt et al., 2017). This can be verified by checking that the distribution $\mathcal{N}(\cdot; w^*, \Sigma)$ satisfies the Fokker-Planck equation (see Sec. D). Having $p_{A_{\text{SGD}}}(w \mid S) = \mathcal{N}(w; w^*, \Sigma)$ and $p_{A_{\text{SGD}}}(w \mid S_{-i}) = \mathcal{N}(w; w^*_{-i}, \Sigma_{-i})$, we have that

$$\text{SI}(z_i, A_{\text{SGD}}) = \frac{1}{2}\left((w^* - w^*_{-i})^T\Sigma^{-1}_{-i}(w^* - w^*_{-i}) + \text{tr}(\Sigma^{-1}_{-i}\Sigma) + \log|\Sigma_{-i}\Sigma^{-1}| - d\right). \tag{21}$$

By the assumption that SGD steady-state covariance stays constant after removing an example, i.e. $\Sigma_{-i} = \Sigma$, equation (21) simplifies to:

$$\text{KL}(p_{A_{\text{SGD}}}(w \mid S) \,\|\, p_{A_{\text{SGD}}}(w \mid S_{-i})) = \frac{1}{2}(w^* - w^*_{-i})^T\Sigma^{-1}(w^* - w^*_{-i}). \tag{22}$$

By the definition of the $A_{\text{ERM}}$ algorithm and smooth sample information, this is exactly equal to $\text{SI}_\Sigma(z_i, A_{\text{ERM}})$. □

## C  SGD NOISE COVARIANCE

Assume we have $n$ examples and the batch size is $b$. Let $g_i \triangleq \nabla_w \mathcal{L}_i(w)$, $g \triangleq \frac{1}{n}\sum_{i=1}^n g_i$, and $\tilde{g} \triangleq \frac{1}{b}\sum_{i=1}^b g_{k_i}$, where $k_i$ are sampled independently from $\{1, \ldots, n\}$ uniformly at. Then

$$\begin{aligned}
\text{Cov}[\tilde{g}, \tilde{g}] &= \mathbb{E}\left[\left(\frac{1}{b}\sum_{i=1}^b g_{k_i} - g\right)\left(\frac{1}{b}\sum_{i=1}^b g_{k_i} - g\right)^T\right] \\
&= \frac{1}{b^2}\sum_{i=1}^b \mathbb{E}\left[(g_{k_i} - g)(g_{k_i} - g)^T\right] \\
&= \frac{1}{bn}\sum_{i=1}^n (g_i - g)(g_i - g)^T \\
&= \frac{1}{b}\left(\frac{1}{n}\left(\sum_{i=1}^n g_i g_i^T\right) - gg^T\right).
\end{aligned}$$

We denote the per-sample covariance, $b \cdot \text{Cov}[\tilde{g}, \tilde{g}]$ with $\Lambda(w)$:

$$\Lambda(w) = \frac{1}{n}\left(\sum_{i=1}^{n} g_i g_i^T\right) - gg^T = \frac{1}{n}GG^T - gg^T,$$

where $G \in \mathbb{R}^{d \times n}$ has $g_i$'s as its columns. We can see that whenever the number of samples times number of outputs is less than number of parameters ($nk < d$), then $\Lambda(w)$ will be rank deficient. Also, note that if we add weight decay to the total loss then covariance $\Lambda(w)$ will not change, as all gradients will be shifted by the same vector.

## D  STEADY-STATE COVARIANCE OF SGD

In this section we verify that the normal distribution $\mathcal{N}(\cdot; w^*, \Sigma)$, with $w^*$ being the global minimum of the regularization MSE loss and covariance matrix $\Sigma$ satisfying the continuous Lyapunov equation $\Sigma H + H\Sigma = \frac{\eta}{b}\Lambda(w^*)$, is the steady-state distribution of the stochastic differential equation of eq. (12). We assume that (a) $\Lambda(w)$ is constant in a small neighborhood of $w^*$ and (b) the steady-state distribution is unique. We start with the Fokker-Planck equation:

$$\frac{\partial p(w,t)}{\partial t} = \sum_{i=1}^{n} \frac{\partial}{\partial w_i}\left[\eta \nabla_{w_i}\mathcal{L}(w)p(w,t)\right] + \frac{\eta^2}{2b}\sum_{i=1}^{n}\sum_{j=1}^{n}\frac{\partial^2}{\partial w_i \partial w_j}\left[\Lambda(w)_{i,j}p(w,t)\right].$$

If $p(w) = \mathcal{N}(w; w^*, \Sigma) = \frac{1}{Z}\exp\left\{-\frac{1}{2}(w-w^*)^T\Sigma^{-1}(w-w^*)\right\}$ is the steady-state distribution, then the Fokker-Planck becomes:

$$0 = \sum_{i=1}^{d} \frac{\partial}{\partial w_i}\left[\eta \nabla_{w_i}\mathcal{L}(w)p(w)\right] + \frac{\eta^2}{2b}\sum_{i=1}^{d}\sum_{j=1}^{d}\frac{\partial^2}{\partial w_i \partial w_j}\left[\Lambda(w)_{i,j}p(w)\right]. \tag{23}$$

In the case of MSE loss:

$$\nabla_w\mathcal{L}(w) = \sum_{k=1}^{n} \nabla f_0(x_k)(f(x_k)-y_k) + \lambda w = \nabla f_0(X)(f(X)-Y) + \lambda w,$$

$$\nabla_w^2\mathcal{L}(w) = \sum_{k=1}^{n} \nabla f_0(x_k) + \lambda I_d \nabla f_0(x_k)^T = \nabla f_0(X)\nabla f_0(X)^T + \lambda I.$$

Additionally, for $p(w)$ the following two statements hold:

$$\frac{\partial}{\partial w_i}p(w) = -p(w)\Sigma_i^{-1}(w-w^*),$$

$$\frac{\partial^2}{\partial w_i \partial w_j}p(w) = -p(w)\Sigma_{i,j}^{-1} + p(w)\Sigma_j^{-1}(w-w^*)\Sigma_i^{-1}(w-w^*),$$

where $\Sigma_i^{-1}$ is the $i$-th row of $\Sigma^{-1}$. Let's compute the first term of (23):

$$\sum_{i=1}^{d} \frac{\partial}{\partial w_i}\left[\nabla_{w_i}\mathcal{L}(w)p(w)\right] = \sum_{i=1}^{d}\left[p(w)\left(\nabla f_0(X)_i\nabla f_0(X)_i^T + \lambda w_i\right) - \nabla_{w_i}\mathcal{L}(w)\cdot p(w)\Sigma_i^{-1}(w-w^*)\right]$$

$$= p(w)\text{tr}\left(\nabla f_0(X)\nabla f_0(X)^T + \lambda I\right) - p(w)\sum_{i=1}^{d}\left(\nabla f_0(X)_i(f(X)-Y) + \lambda w_i\right)\Sigma_i^{-1}(w-w^*)$$

$$= p(w)\text{tr}(H) - p(w)\left((f(X)-Y)^T\nabla f_0(X)^T + \lambda w^T\right)\Sigma^{-1}(w-w^*). \tag{24}$$

As $w^*$ is a critical point of $\mathcal{L}(w)$, we have that $\nabla f_0(X)(f_{w^*}(X)-Y) + \lambda w^* = 0$. Therefore, we can subtract $p(w)\left((f_{w^*}(X)-Y)^T\nabla f_0(X)^T + \lambda(w^*)^T\right)\Sigma^{-1}(w-w^*)$ from (24):

$$\sum_{i=1}^{d}\frac{\partial}{\partial w_i}\left[\nabla_{w_i}\mathcal{L}(w)p(w)\right] =$$

$$= p(w)\text{tr}(H) - p(w)\left((f(X)-f_{w^*}(X))^T\nabla f_0(X)^T + \lambda(w-w^*)^T\right)\Sigma^{-1}(w-w^*)$$

$$= p(w)\text{tr}(H) - p(w)(w-w^*)^T\left(\nabla f_0(X)\nabla f_0(X)^T + \lambda I\right)\Sigma^{-1}(w-w^*)$$

$$= p(w)\text{tr}(H) - p(w)(w-w^*)^T H\Sigma^{-1}(w-w^*). \tag{25}$$

ISOTROPIC CASE: $\Lambda(w) = \sigma^2 I_d$

In the case when $\Lambda(w) = \sigma^2 I_d$, we have

$$\sum_{i,j} \frac{\partial^2}{\partial w_i \partial w_j} [\Lambda(w)_{i,j} p(w)] = \sigma^2 \text{tr}(\nabla_w^2 p(w)) = -\sigma^2 p(w) \text{tr}(\Sigma^{-1}) + \sigma^2 p(w) \|\Sigma^{-1}(w - w^*)\|_2^2.$$

Putting everything together in the Fokker-Planck we get:

$$\eta \left( p(w) \text{tr}(H) - p(w)(w - w^*)^T H \Sigma^{-1}(w - w^*) \right)$$
$$+ \frac{\eta^2}{2b} \left( -\sigma^2 p(w) \text{tr}(\Sigma^{-1}) + \sigma^2 p(w) \|\Sigma^{-1}(w - w^*)\|_2^2 \right) = 0.$$

It is easy to verify that $\Sigma^{-1} = \frac{2b}{\eta\sigma^2} H$ is a valid inverse covariance matrix and satisfies the equation above. Hence, it is the unique steady-state distribution of the stochastic differential equation. The result confirms that variance is high when batch size is low or learning rate is large. Additionally, the variance is low along directions of low curvature.

NON-ISOTROPIC CASE

We assume $\Lambda(w)$ is constant around $w^*$ and is equal to $\Lambda$. This assumption is acceptable to a degree, because SGD converges to a relatively small neighborhood, in which we can assume $\Lambda(w)$ to not change much. With this assumption,

$$\sum_{i,j} \frac{\partial^2}{\partial w_i \partial w_j} [\Lambda_{i,j} p(w)] = \sum_{i,j} \Lambda_{i,j} \left[ -p(w)\Sigma_{i,j}^{-1} + p(w)\Sigma_j^{-1}(w - w^*)\Sigma_i^{-1}(w - w^*) \right]$$
$$= -p(w)\text{tr}(\Sigma^{-1}\Lambda) + p(w) \sum_{i,j} \Lambda_{i,j} (\Sigma^{-1}(w - w^*)(w - w^*)^T \Sigma^{-1})_{i,j}$$
$$= -p(w)\text{tr}(\Sigma^{-1}\Lambda) + p(w)\text{tr}(\Sigma^{-1}(w - w^*)(w - w^*)^T \Sigma^{-1}\Lambda)$$
$$= -p(w)\text{tr}(\Sigma^{-1}\Lambda) + p(w)(w - w^*)^T \Sigma^{-1}\Lambda\Sigma^{-1}(w - w^*)). \tag{26}$$

It is easy to verify that if $\Sigma H + H\Sigma = \frac{\eta}{b}\Lambda$, then terms in equations (25) and (26) will be negatives of each other up to a constant $\frac{\eta}{2b}$, implying that $p(w)$ satisfies the Fokker-Planck equation. Note that $\Sigma^{-1} = \frac{2b}{\eta} H\Lambda^{-1}$ also satisfies the Fokker-Planck, but will not be positive definite unless $H$ and $\Lambda$ commute.

## E    FAST UPDATE OF NTK INVERSE AFTER DATA REMOVAL

For computing weights or predictions of a linearized network at some time $t$, we need to compute the inverse of the NTK matrix. To compute the informativeness scores, we need to do this inversion $n$ time, each time with one data point excluded. In this section, we describe how to update the inverse of NTK matrix after removing one example in $O(n^2 k^3)$ time, instead of doing the straightforward $O(n^3 k^3)$ computation. Without loss of generality let's assume we remove last $r$ rows and corresponding columns from the NTK matrix. We can represent the NTK matrix as a block matrix:

$$\Theta_0 = \begin{bmatrix} A_{11} & A_{12} \\ A_{21} & A_{22} \end{bmatrix}.$$

The goal is to compute $A_{11}^{-1}$ from $\Theta_0^{-1}$. We start with the block matrix inverse formula:

$$\Theta_0^{-1} = \begin{bmatrix} A_{11} & A_{12} \\ A_{21} & A_{22} \end{bmatrix}^{-1} = \begin{bmatrix} F_{11}^{-1} & -F_{11}^{-1} A_{12} A_{22}^{-1} \\ -A_{22}^{-1} A_{21} F_{11}^{-1} & F_{22}^{-1} \end{bmatrix}, \tag{27}$$

where

$$F_{11} = A_{11} - A_{12} A_{22}^{-1} A_{21}, \tag{28}$$
$$F_{22} = A_{22} - A_{21} A_{11}^{-1} A_{12}. \tag{29}$$

From (28) we have $A_{11} = F_{11} + A_{12}A_{22}^{-1}A_{21}$. Applying the Woodbury matrix identity on this we get:

$$A_{11}^{-1} = F_{11}^{-1} - F_{11}^{-1}A_{12}(A_{22} + A_{21}F_{11}^{-1}A_{12})^{-1}A_{21}F_{11}^{-1}. \tag{30}$$

This Eq. (30) gives the recipe for computing $A_{11}^{-1}$. Note that $F_{11}^{-1}$ can be read from $\Theta_0^{-1}$ using (27), $A_{12}, A_{21}$, and $A_{22}$ can be read from $\Theta$. Finally, the complexity of computing $A_{11}^{-1}$ using (30) is $O(n^2k^3)$ if we remove one example.

# F  ADDING WEIGHT DECAY TO LINEARIZED NEURAL NETWORK TRAINING

Let us consider the loss function $\mathcal{L}(w) = \sum_{i=1}^n \mathcal{L}_i(w) + \frac{\lambda}{2}\|w - w_0\|_2^2$. In this case continuous-time gradient descent is described by the following ODE:

$$\dot{w}(t) = -\eta\nabla_w f_0(X)(f_t(X) - Y) - \eta\lambda(w(t) - w_0) \tag{31}$$
$$= -\eta\nabla_w f_0(X)(\nabla_w f_0(X)^T(w(t) - w_0) + f_0(X) - Y) - \eta\lambda(w(t) - w_0) \tag{32}$$
$$= \underbrace{-\eta(\nabla_w f_0(X)\nabla_w f_0(X)^T + \lambda I)}_{A}(w(t) - w_0) + \underbrace{\eta\nabla_w f_0(X)(-f_0(X) + Y)}_{b}. \tag{33}$$

Let $\omega(t) \triangleq w(t) - w_0$, then we have

$$\dot{\omega}(t) = A\omega(t) + b. \tag{34}$$

Since all eigenvalues of $A$ are negative, this ODE is stable and has steady-state

$$\omega^* = -A^{-1}b \tag{35}$$
$$= (\nabla_w f_0(X)\nabla_w f_0(X)^T + \lambda I)^{-1}\nabla_w f_0(X)(Y - f_0(X)). \tag{36}$$

The solution $\omega(t)$ is given by:

$$\omega(t) = \omega^* + e^{At}(\omega_0 - \omega^*) \tag{37}$$
$$= (I - e^{At})\omega^*. \tag{38}$$

Let $\Theta_w \triangleq \nabla_w f_0(X)\nabla_w f_0(X)^T$ and $\Theta_0 \triangleq \nabla_w f_0(X)^T\nabla_w f_0(X)$. If the SVD of $\nabla_w f_0(X)$ is $UDV^T$, then $\Theta_w = UDU^T$ and $\Theta_0 = VDV^T$. Additionally, we can extend the columns of $U$ to full basis of $\mathbb{R}^d$ (denoted with $\tilde{U}$) and append zeros to $D$ (denoted with $\tilde{D}$) to write down the eigen decomposition $\Theta_w = \tilde{U}\tilde{D}\tilde{U}^T$. With this, we have $(\Theta_w + \lambda I)^{-1} = \tilde{U}(\tilde{D} + \lambda I)^{-1}\tilde{U}^T$. Continuing (38) we have

$$\omega(t) = (I - e^{At})\omega^* \tag{39}$$
$$= (I - e^{At})(\Theta_w + \lambda I)^{-1}\nabla_w f_0(X)(Y - f_0(X)) \tag{40}$$
$$= (I - e^{At})\tilde{U}(\tilde{D} + \lambda I)^{-1}\tilde{U}^T UDV^T(Y - f_0(X)) \tag{41}$$
$$= \tilde{U}(I - e^{-\eta t(\tilde{D} + \lambda I)})\tilde{U}^T\tilde{U}(\tilde{D} + \lambda I)^{-1}\tilde{U}^T UDV^T(Y - f_0(X)) \tag{42}$$
$$= \tilde{U}(I - e^{-\eta t(\tilde{D} + \lambda I)})(\tilde{D} + \lambda I)^{-1}I_{d\times nk}DV^T(Y - f_0(X)) \tag{43}$$
$$= \tilde{U}(I - e^{-\eta t(\tilde{D} + \lambda I)})\tilde{Z}V^T(Y - f_0(X)), \tag{44}$$

where $\tilde{Z} = \begin{bmatrix} (D + \lambda I)^{-1}D \\ 0 \end{bmatrix} \in \mathbb{R}^{d\times nk}$. Denoting $Z \triangleq (D + \lambda I)^{-1}D$ and continuing,

$$\omega(t) = \tilde{U}(I - e^{-\eta t(\tilde{D} + \lambda I)})\tilde{Z}V^T(Y - f_0(X)) \tag{45}$$
$$= U(I - e^{-\eta t(D + \lambda I)})ZV^T(Y - f_0(X)) \tag{46}$$
$$= UZ(I - e^{-\eta t(D + \lambda I)})V^T(Y - f_0(X)) \tag{47}$$
$$= UZV^TV(I - e^{-\eta t(D + \lambda I)})V^T(Y - f_0(X)) \tag{48}$$
$$= UZV^T(I - e^{-\eta t(\Theta_0 + \lambda I)})(Y - f_0(X)) \tag{49}$$
$$= \nabla_w f_0(X)(\Theta_0 + \lambda I)^{-1}(I - e^{-\eta t(\Theta_0 + \lambda I)})(Y - f_0(X)). \tag{50}$$

**Solving for outputs.** Having $w(t)$ derived, we can write down dynamics of $f_t(x)$ for any $x$:

$$f_t(x) = f_0(x) + \nabla_w f_0(x)^T \omega(t) \tag{51}$$

$$= f_0(x) + \nabla_w f_0(x)^T \nabla_w f_0(X)(\Theta_0 + \lambda I)^{-1}(I - e^{-\eta t(\Theta_0 + \lambda I)})(Y - f_0(X)) \tag{52}$$

$$= f_0(x) + \Theta_0(x, X)(\Theta_0 + \lambda I)^{-1}(I - e^{-\eta t(\Theta_0 + \lambda I)})(Y - f_0(X)). \tag{53}$$

