# OpenReview forum: "Estimating informativeness of samples with Smooth Unique Information"
_ICLR.cc/2021/Conference — ICLR 2021 Poster_

### Official Review · AnonReviewer1 · 2020-10-28

**Rating:** 6
**Confidence:** 4

**Review:**

The paper proposes a measure to compute the information of each training sample. The paper shows that this measure can be computed for a large DNN without having to train the network. The authors show that this measure can be used for applications like data summarization and detection of corrupted data.

My main concern is that the information measure depends on the initialization, training time, and network architecture. For e.g. the F-SI scores seem to change over time in table 5. Hence, one needs to re-compute the measure for the training samples, with a  change in initialization and training time. The complexity of computing the FI-scores for $m$ examples is $O(n^2 m)$. For MNIST, this is at least $10^{10}$ flops for computing measure of 100 examples. I believe, the authors should discuss efficient recomputation of the measure in case of a change in the algorithm parameters or the error involved if we don't re-compute the measure.


My other concerns are the following:
1. The data summarization experiments and the detection of more information data sources and examples are network-specific (in this case, pre-trained Resnet 18). It will be more interesting to see if these transfer to other networks. For example, in data summarization experiments, it will be interesting to see if the most informative examples for pre-trained Resnet 18 also help to train another network. The same question holds for SVHN vs MNIST.
2. In conclusion, the authors claim that their measure can be used for computing the information for a group of samples. However, in the data summarization experiments, the measure is computed for each sample as a separate entity, while groups of samples are removed at a time. Hence, is it possible to compare the current results with an efficient group-theoretic measure for removing groups of samples? I am concerned because simple examples may not have information individually but as a group may hold important information to train a network.
3. In many sections, the authors have forgotten to mention the network architecture they used to get the plots e.g. in data summarization and detection of under sampled sub-classes experiments. It will be great if the authors can show how the results in these experiments change with a change in architecture, given that the measure depends on the architecture.

I have verified the proofs. They are easy to read and understand. My scores are slightly on the lower side because I believe the computation is too brittle to changes in the algorithm parameters like training time and initialization. I am happy to discuss this with the authors and other reviewers during the discussion period.


***After Rebuttal***
I have read the reviews by other reviewers and the responses of the authors to the questions posed by other reviewers. I enjoyed the additional experiments that the authors added to the paper during the rebuttal. The paper has shown interesting observations on the informative samples present in real-world datasets for different architectures. However, I still believe the method proposed by the paper is too inefficient for simple algorithm changes like changes in initialization. Hence, I am keeping the score the same after the rebuttal.

---

> ### Author Response · Authors · 2020-11-19
> **Response (part 1)**
>
> We thank the reviewer for the comments and suggestions.
>
> >My main concern is that the information measure depends on the initialization, training time, and network architecture. For e.g. the F-SI scores seem to change over time in table 5. Hence, one needs to re-compute the measure for the training samples, with a change in initialization and training time. The complexity of computing the FI-scores for m examples is  O(n^2m). For MNIST, this is at least  10^10 flops for computing measure of 100 examples. I believe, the authors should discuss efficient recomputation of the measure in case of a change in the algorithm parameters or the error involved if we don't re-compute the measure.
>
> Indeed, the information measures depend on initialization, training time and network (i.e., on the training algorithm). This has to be the case, as an example can be informative with respect to one algorithm, or one architecture, but not for another one (e.g., an informative sample for a CNN does not need to be informative for a linear classifier). Nevertheless, we expect the information score to be robust when using similar algorithms. Indeed, we found that there is significant correlation between sample information computed for different initializations, architectures, and training lengths. We provide additional evidence of this in the revised Sec. A4 of the supplementary material (see also the response below for additional discussion).
>
> Changing the initialization or the network changes the NTK matrix (although we expect the new matrix to be similar), therefore recomputation from scratch is the only way. When changing the training length, we can reuse the NTK matrix (which is otherwise the most time-consuming step), its inverse, and its SVD. In practice, this is much more efficient than computing from scratch. Additionally, we would like to note it takes only about 2 minutes on a GTX 2080Ti GPU to compute both SI and F-SI for all examples from scratch on MNIST with $nk=1000$.
>
> >The data summarization experiments and the detection of more information data sources and examples are network-specific (in this case, pre-trained Resnet 18). It will be more interesting to see if these transfer to other networks. For example, in data summarization experiments, it will be interesting to see if the most informative examples for pre-trained Resnet 18 also help to train another network. The same question holds for SVHN vs MNIST.
>
> In the appendix we compute informativeness scores with ResNet-18 and ResNet-50, and find that there is around 34% correlation. We also add new results where we compute correlations with a completely different architecture, DenseNet-121. We found that correlation between F-SIs computed for ResNet-50 and DenseNet-121 is around 45%. In particular, note that wider architectures (ResNet-50 and DenseNet-121) tend to agree more than smaller architectures (ResNet-18). This is expected since in the limit of a wide network the training dynamics are expected to converge to the same NTK limit. Moreover, when we plot the top 10 most informative images for ResNet-18, ResNet-50, DenseNet-121, we see that several images are shared between them. These new results are added to the supplementary material.
>
> As suggested, we also perform a new data summarization experiment. We computed informativeness scores with respect to the original one-hidden-layer network, but then did the training with a two-hidden-layer network. We found that the resulting data summarization plot is qualitatively and quantitatively almost identical to the one presented in the paper. This confirms that informativeness scores computed for one network can also be useful for another network. We have added this new result to the appendix.

---

> > ### Comment · AnonReviewer1 · 2020-11-23
> > **Thanks for your response**
> >
> > Thanks for your quick response. I have read the revised version and your response to the queries raised by other reviewers. I feel there are a lot of interesting additions to the revised version, one of them being the experiments on sentiment analysis.
> >
> > I still have my concern about the complexity of computing the F-SI scores. Computing the F-SI scores takes approximately 2 minutes for MNIST data when $nk  = 1000$. That would imply it takes around $1000$ minutes to compute the F-SI scores for $100$ examples in the entire MNIST dataset, which has $n=50000$ training examples. I still feel this is a bit expensive, given that we need to re-compute for a slight change in network parameters like initialization.

---

> > > ### Author Response · Authors · 2020-11-24
> > > **Thanks for the reply**
> > >
> > > Indeed computational complexity is a challenge, which we plan to address in future work.
> > > We focused on a method which is efficient to run on smaller datasets, since it is there that the effect of individual data is most pronounced. Such small datasets, where computing a measure of informativeness of individual samples is meaningful, often arise in active learning and transfer learning. In this setting, our method takes minutes while competing methods, such as Influence Functions, take hours. For larger datasets ($nk \ge 20000$), we recommend using a conjugate-gradient-based method in weight space (which scales linearly with the number of parameters) instead of the NTK-based method in function space (which scales quadratically with the number of training samples), to compute the proposed functional sample information measure.

---

> > > > ### Comment · AnonReviewer1 · 2020-11-24
> > > > **Thanks for your response**
> > > >
> > > > This comparison gives me some insight into the computational complexity of your algorithm compared to the previous ones. It will be good to have a table on this comparison, and also a discussion on the conjugate-gradient-based method.

---

> ### Author Response · Authors · 2020-11-19
> **Response (part 2)**
>
> >In conclusion, the authors claim that their measure can be used for computing the information for a group of samples. However, in the data summarization experiments, the measure is computed for each sample as a separate entity, while groups of samples are removed at a time. Hence, is it possible to compare the current results with an efficient group-theoretic measure for removing groups of samples? I am concerned because simple examples may not have information individually but as a group may hold important information to train a network.
>
> The non-iterative approach (the “bottom” baseline) is the greediest approach and ignores the fact that examples in a group can be non-informative individually, but informative as a group. For this reason, we also consider the “bottom-iterative” approach, which addresses this problem – albeit not entirely – by recomputing the information measures after each removal. This way we can account for the fact that removing a sample can increase the informativeness of another. To completely follow the group-theoretic approach in a non-greedy manner, we need to find the least/most informative group of k examples. This is computationally challenging to do exactly due to the combinatorial nature of the problem.
>
> >In many sections, the authors have forgotten to mention the network architecture they used to get the plots e.g. in data summarization and detection of under sampled sub-classes experiments. It will be great if the authors can show how the results in these experiments change with a change in architecture, given that the measure depends on the architecture.
>
> We added the missing information about which network architectures were used. The experiment of detecting mislabeled examples was done for different architectures and datasets (Fig. 7). The results were qualitatively similar to those already reported in the paper. We tried using a pretrained DenseNet-121 in the MNIST vs SVHN experiment. We found the results to be qualitatively identical. We have added these results to the appendix.

---

### Official Review · AnonReviewer2 · 2020-10-29
**A new way to address an old issue with interesting theoretical insights but unclear practical merits**

**Rating:** 6
**Confidence:** 4

**Review:**

The paper proposes a way to estimating sample information in the
context of neural networks.  The authors propose to simplify the definition
by using a first-order approximation of an arbitrary network's architecture
and the mean squared error as the loss function.  In addition, a smoothing
matrix obtained around the network's steady state is introduced to minimize the
uncertainty due to limited realizations of a stochastic optimization process.
The method is applied to several image classification tasks for illustration.
Then it is proposed to be used to summarize a dataset, i.e., to remove redundant
examples that have minimal impact to training results.

The derivation of the measure is an interesting exploration of the
relationship between an individual sample's information content and
the neural network's final trained weights or the network's use of
such weights (the values of the decision function).  This contributes
to a better understanding of how information is leveraged by neural networks.

The practical value of the measure itself, on the other hand, is
questionable beyond the use of neural networks.

Is there a way to characterize how intrinsic this measure is,
i.e., to what extent is it tied to the use of neural networks as
the classifier?  Note that there are much simpler, well studied ways
to estimate how normal or abnormal a sample is compared to the rest in
its class and how much it affects classification difficulty
(see the above mentioned survey).  Going through these complicated
calculations for estimation, do you arrive at something that well
correlates with many others derivable using simpler methods?

You may want to compare your data summary to what could be obtained
using, e.g., "The Condensed Nearest Neighbor Rule", Peter Hart, IEEE
Transactions on Information Theory, May 1968, 515-516.

The illustrative examples are taken only from image classification.
It will be more convincing if other tasks, e.g., text classification,
are also tried.  Even better, you can show the values of the
measure, and what outliers it can pull out on some extreme cases:
e.g. randomly labeled samples, or perfectly separable, synthetic
classes.

What would the measure say for the adversarial samples crafted to fool
a neural network?


Misc.:

Section 2, related works:  should also refer to a recent survey that
includes many works analyzing the influence of individual samples
on classification difficulty:
"How complex is your classification problem? A survey on measuring
classification complexity", by AC Lorena, et al., ACM Computing
Surveys, 52(5), 1-34.

p.5:  in and before Proposition 5.1:  something is missing in what is
referred to as "SGD's steady-state covariance".  SGD usually refers to
the optimization procedure; are you reusing it to denote the dw quantity?
If not, what is the quantity whose distribution this covariance is about?

Figure 1B:  why only samples of class cmd are shown?
To make a better illustration, you may want to show both informative
and non-informative samples for at least two classes.  Also, what do
the histograms in Figure 1A tell you about the different classes?
The examples from Figure 4 may serve the purpose of illustration
better as they are more familiar objects.

---

> ### Author Response · Authors · 2020-11-19
> **Response (part 1)**
>
> We thank the reviewer for the comments and suggestions.
>
> >The practical value of the measure itself, on the other hand, is questionable beyond the use of neural networks.
>
> The proposed definitions of sample information apply to any parametric learning algorithm. On the other hand, the proposed method of estimation (i.e., the linearization) is indeed specific to neural networks. We note that the proposed method scales more easily to large networks than competing methods based on similar approximations, such as influence functions.
>
>
>
> >Is there a way to characterize how intrinsic this measure is, i.e., to what extent is it tied to the use of neural networks as the classifier? Note that there are much simpler, well studied ways to estimate how normal or abnormal a sample is compared to the rest in its class and how much it affects classification difficulty (see the above mentioned survey). Going through these complicated calculations for estimation, do you arrive at something that well correlates with many others derivable using simpler methods?
> You may want to compare your data summary to what could be obtained using, e.g., "The Condensed Nearest Neighbor Rule", Peter Hart, IEEE Transactions on Information Theory, May 1968, 515-516.
>
> Indeed, there is a large literature on characterizing abnormal or out-of-distribution samples. However, our main goal in this paper is to measure how informative a training sample is to a given machine learning model and training algorithm, which motivates our derivation. In particular, abnormal samples are not necessarily more or less informative. For example, in Fig. 2a both MNIST and SVHN examples are typical in their respective distributions, but SVHN samples are on average more informative.
>
> On the other hand, we observe that some samples are considered more informative regardless of the particular architecture, that is, that informativeness is intrinsic to some extent. As also mentioned in the response to R1, in Sec. A4 we have added more experiments to show that different architectures (e.g., ResNet-50 and DenseNet-101) and training algorithms have a good agreement on which samples are more informative.
>
> >The illustrative examples are taken only from image classification. It will be more convincing if other tasks, e.g., text classification, are also tried. Even better, you can show the values of the measure, and what outliers it can pull out on some extreme cases: e.g. randomly labeled samples, or perfectly separable, synthetic classes.
> What would the measure say for the adversarial samples crafted to fool a neural network?
>
> As the reviewer suggested, we added a new experiment, involving text classification. We consider a sentiment analysis task, with 2000 samples from IMDB reviews dataset. We use a pretrained GloVe embedding and convert each review to a 300-dimensional vector by taking the mean of the embeddings of all words. Then we apply a neural network on top of this, with one hidden layer, consisting of 2000 ReLU units. Finally, we compute the F-SI for all training examples, and rank them. Our analysis shows that the least informative examples are the ones which unambiguously reveal the sentiment, by using sentiment-specific adjectives many times.
> On the other hand, we found the most informative examples to be trickier and more unusual. For example, the following two reviews are among most informative ones:
>
> > “smallville episode justice is the best episode of smallville ! ! ! ! ! ! ! ! ! ! ! ! ! ! ! ! ! ! ! ! ! ! ! ! ! ! ! ! ! ! ! ! ! ! ! ! ! ! ! ! ! ! ! ! ! ! ! ! ! ! ! ! ! ! ! ! ! ! ! ! ! ! ! ! ! ! ! ! ! ! ! ! ! ! ! ! ! ! ! ! ! ! ! ! ! ! ! ! ! ! ! ! ! ! ! ! ! ! ! ! ! ! ! ! ! ! ! ! ! ! ! ! ! ! ! ! ! ! ! ! ! ! ! ! ! ! ! ! ! ! ! ! ! ! ! ! ! ! ! ! ! ! ! ! it's my favorite episode of smallville! ! ! ! ! ! ! ! ! ! ! ! ! ! ! ! ! ! ! ! ! ! ! ! ! ! ! ! ! ! ! ! ! ! ! ! ! ! ! ! ! ! ! ! ! ! ! ! ! ! ! ! ! ! ! ! ! ! ! ! ! ! ! ! ! ! ! ! ! ! ! ! ! ! ! ! ! ! ! ! ! ! ! ! ! ! ! ! ! ! ! ! ! ! ! ! ! ! ! ! ! ! ! ! ! ! ! ! ! ! ! ! ! ! ! ! ! ! ! ! ! ! ! ! ! ! ! ! ! ! ! ! ! ! ! ! ! !”
>
> >”i thought this movie would be dumb, but i really liked it. people i know hate it because spirit was the only horse that talked. well, so what? the songs were good, and the horses didn't need to talk to seem human. i wouldn't care to own the movie, and i would love to see it again.”
>
> Furthermore, we found the least informative examples to be on average longer than the most informative ones, possibly because averaging many word vectors makes long reviews similar to each other.
>
> As per randomly labeled examples, the figures 2b and 6 show that F-SI can detect them.

---

> ### Author Response · Authors · 2020-11-19
> **Response (part 2)**
>
> We tried computing F-SI scores with respect to a linear regression applied to the almost perfectly separable synthetic 2D dataset shown in the figure 3. We observe that all examples get close to zero unique information scores. This is expected as no single example in this dataset has a large effect on the resulting linear regressor. However, if we then add 100 extra noise dimensions to the data, we observe that examples that are close to the decision boundary start to be ranked as more informative.
>
> Regarding adversarial examples, we added a new experiment. We consider the Kaggle cats vs dogs classification task with 1000 samples, where the network is a pretrained ResNet-18. After fine-tuning the network on this dataset, for 10% of the examples we create successful adversarial images using the FGSM with eps=0.01. Then for those 10% of the examples we replace the original images with the corresponding adversarial images and consider a new pretrained ResNet-18. For this new network we compute the F-SI scores and find that, on average, adversarial examples are much more informative than normal ones. This is in agreement with the fact that adding adversarial examples to the training process (adversarial training) produces a more robust model, and hence that adversarial examples provide additional information to the training process.
>
> >p.5: in and before Proposition 5.1: something is missing in what is referred to as "SGD's steady-state covariance". SGD usually refers to the optimization procedure; are you reusing it to denote the dw quantity? If not, what is the quantity whose distribution this covariance is about?
>
> “SGD’s steady-state covariance” is a short-hand for “the covariance of the steady-state distribution of the stochastic differential equation (12), which describes the continuous-time SGD”. In other words, it is the covariance of the random variable $w_t$ in the limit of $t \to \infty$.
>
> >Figure 1B: why only samples of class cmd are shown?
> To make a better illustration, you may want to show both informative and non-informative samples for at least two classes. Also, what do the histograms in Figure 1A tell you about the different classes? The examples from Figure 4 may serve the purpose of illustration better as they are more familiar objects.
>
> Figure 1B only shows examples from the “cmd” class because all top 10 least informative examples were from that class. The histograms in Figure 1A tell us that examples from different classes have on average very different extents of informativeness. As we discuss in the last section of that paragraph, this is connected to class imbalance (in fact, the average unique information of the samples in one class depends on several factors, such as the variability of the class, the amount of samples, and how easy it is to discriminate). In the main text, we choose to show examples from iCassava datasets rather than from MNIST or Cats vs Dogs, partly because of the findings of Figure 1A.

---

### Official Review · AnonReviewer4 · 2020-10-29
**Very weak presentation. Significant overhaul needed.**

**Rating:** 6
**Confidence:** 3

**Review:**

The paper is written in a very bad way and to my opinion is not acceptable without a significant overhaul. First, there are many typos. For instance, word out in page 1 should be our. Secondly, and much more importantly, notations in the paper are very ambiguous and misleading. For instance, in the "prerequisites and notations" section it is mentioned that $A(w|S)$ is a "conditional distribution". Yet in the same section, it says that $A(S)$ is the "output random variable" of the algorithm. So it is absolutely ambiguous what the notation $A(.)$ actually shows. Does it show a probability distribution? or it shows a random variable? You would guess that if you keep reading the paper, the ambiguity goes away, but you are wrong. As you read through the paper, $A(.)$ is used interchangeably for both a random variable and a probability distribution which absolutely ambiguous. For instance, when you see $KL(A(w | S) || A(w | S_{−i}))$ in equation 3 you would argue that $A(w|S)$ refers to a probability distribution that in Prop. 3.2. you would see that $A(S)$ is used as a random variable. Why are the authors use notation $A(.)$ for both a random variable and a probability distribution. I think to remove ambiguity, all probability distributions should be shown with $P(.), p(.), f(.)$ or things of that nature. I would expect to see this fixed in any future revision of the paper. Also, notations like $m(.)$ are not common for probability distributions and make the paper unreadable.

The role of smoothing in the paper is not clearly discussed and analyzed in the paper. I understand that it is required to make the KL divergence bounds work by adding continuous noise, but are the authors assume assumptions like this just to get some theoretical bounds? Is such an assumption, a valid assumption? and why? How does it change the results compared to real-world applications?

---

> ### Author Response · Authors · 2020-11-19
> **Response**
>
> We thank the reviewer for the comments and suggestions.
>
> We have corrected the typos in our updated paper. Regarding the use of the symbol $A$: we denote with $w = A(S)$ is the output of the learning algorithm $A$, which is a stochastic function (thus $w$ is a random variable). Since the output $w$ is stochastic, it has an associated distribution conditioned on the input S of the training algorithm. We denote this conditional distribution with $A(w | S)$. This overloads the letter ‘A’, but this was intended to keep the notation intuitive. We have both $A(S)$ and $A(w | S)$ since in some cases it is convenient to use the $A(S)$ notation (e.g., when doing smoothing) and in some other cases it is convenient to use the $A(w | S)$ notation (e.g., inside KL divergences). We added a clarification on this notation in the “prerequisites and notations” section. If the reviewer still thinks that this notation is potentially confusing, we can replace $A(w | S)$ with $p_A(w | S)$ to further emphasize that is the probability distribution corresponding to the output of the training algorithm A.
>
> > Also, notations like  m(.)  are not common for probability distributions and make the paper unreadable.
>
> We used $m$ to denote the “[marginal] distribution of the weights over all possible sampling of $z_i$”. We replaced it with the symbol $r$ in the updated paper to eliminate possible sources of confusion.
>
> >The role of smoothing in the paper is not clearly discussed and analyzed in the paper. I understand that it is required to make the KL divergence bounds work by adding continuous noise, but are the authors assume assumptions like this just to get some theoretical bounds? Is such an assumption, a valid assumption? and why? How does it change the results compared to real-world applications?
>
> Indeed, one advantage of using smoothing to define information is that it makes all quantities well-defined even when the training algorithm is discrete. Note however that this is *not* an assumption, the smoothing is part of our definition of Smooth Unique Information (Definition 3.1). From the theoretical perspective, the proposed definitions are useful, as they connect to each other, to classical stability results, and to non-smoothed unique information (in some special cases). Furthermore, smoothing with different choices of $\Sigma$ also allows us to study different ways in which a sample can be informative. Using $\Sigma$ equal to the identity matrix allows us to measure how informative the sample is for the final weights. On the other hand, as we discussed in Section 4, taking Sigma to be the Fisher Information Matrix allows us to measure how informative a sample is for the predictions of the network. That is, it measures whether removing a given sample to the training set will significantly change the test prediction of the trained network. Since for an over-parameterized model like a DNN there is a large subspace of weights that give the same predictions, the two measures can be very different. We have shown the practical value of proposed measures  in tasks such as data summarization without training, detecting mislabeled or (in the updated paper) adversarial examples, and, in general, getting insights about the dataset.

---

> > ### Comment · AnonReviewer4 · 2020-11-21
> > **Presentation needs work**
> >
> > The presentation still needs a lot of work. Not much has changed. Yes, please change $A(w|S)$ to $P_A(w|S)$ to make it distinguishable from $A(S)$ which is the output of the algorithm and is a random variable. Clearly mention in the paper what is a random variable and what is a probability distribution.
> >
> > Also, $r(.)$ is defined as  $r(w | S_{−i} = S_{−i}) = E_{z_i^{\prime} \sim p(z)}[A(w | S = S_{−i}, z_i^{\prime})]$. This, based on what you have added in blue in the Prerequisites and Notation section is an expectation over a conditional distribution function. I can understand the operation of expectation over a random variable but not over a distribution function. Is $A(w | S = S_{−i}, z_i^{\prime})$ a r.v. or a function? It appears to me a function based on your explanations in Prerequisites and Notation section but expectation suggests otherwise. Please clarify.
> >
> > Also, $r(.)$ is not a great alternative either both are equally confusing. If this is a probability density function you will need to represent it by $p(.)$ or $P(.)$.
> >
> > The reference to (Shwartz-Ziv & Tishby, 2017) should be removed as this is a highly contentious paper with flawed results as shown in: Saxe, Andrew M., et al. "On the information bottleneck theory of deep learning." Journal of Statistical Mechanics: Theory and Experiment 2019.12 (2019): 124020.

---

> > > ### Author Response · Authors · 2020-11-24
> > > **Thanks for the reply**
> > >
> > > >The presentation still needs a lot of work. Not much has changed.
> > >
> > > We have addressed all the concerns the reviewer has identified. If there are any additional specific concerns, we would be happy to address them.
> > >
> > >
> > > > Yes, please change $A(w|S)$ to $P_A(w|S)$ to make it distinguishable from $A(S)$ which is the output of the algorithm and is a random variable. Clearly mention in the paper what is a random variable and what is a probability distribution.
> > >
> > > We have made a change in the most recent revision that indicates under “Prerequisites and Notation” that $A(S)$ is the output of a stochastic function (a random variable), and $p_A(w|S)$ is its conditional distribution.
> > >
> > > >Also, $r(.)$ is defined as $r(w|S_{−i}=S_{−i})=\mathbb{E}_{z_i′ \sim p(z)}[A(w|S=S_{−i}, z_i′)]$. This, based on what you have added in blue in the Prerequisites and Notation section is an expectation over a conditional distribution function. I can understand the operation of expectation over a random variable but not over a distribution function. Is $A(w|S=S_{−i}, z_i′)$ a r.v. or a function? It appears to me a function based on your explanations in Prerequisites and Notation section but expectation suggests otherwise. Please clarify.
> > > Also, r(.) is not a great alternative either both are equally confusing. If this is a probability density function you will need to represent it by p(.) or P(.).
> > >
> > > Given a conditional distribution $p(y|x)$ we can write the marginal distribution of $y$ as $p(y) = \int p(y|x) d P(x) = E_x[p(y|x)] $. Writing the marginal using the expectation is used both in popular references (e.g.,  https://en.wikipedia.org/w/index.php?title=Marginal_distribution&oldid=982342411) and in textbooks (e.g., Theoretical Statistics, Keener, Section 6.4). In our case, we are defining the marginal $r(w | S_{-i}) = \int p_A(w| S=S_{-i}, z_i’) dP(z_i’) = E_{z_i} [ p_A(w| S=S_{-i}, z_i’)]$. Note that we can take the expectation of any (positive or integrable) measurable function, and $p_A(w| S=S_{-i}, z_i’)$ is a measurable function of $z_i’$ for fixed $w$ and $S_{-i}$. We have added the notation with the integral to further avoid any confusion.
> > >
> > > > The reference to (Shwartz-Ziv & Tishby, 2017) should be removed as this is a highly contentious paper with flawed results as shown in: Saxe, Andrew M., et al. "On the information bottleneck theory of deep learning." Journal of Statistical Mechanics: Theory and Experiment 2019.12 (2019): 124020.
> > >
> > > We cite (Shwartz-Ziv and Tishby, 2017) only in reference to the sentence “Other complementary works study how information about an input sample propagates through the network”, since it has been influential in suggesting to look at the amount of Shannon Information that each layer has about the input. We do not reference, nor build upon, the empirical results disputed by Saxe et al. that networks trained with SGD show a compression phase. We added a reference to Saxe et al. for completeness.

---

> > > > ### Comment · AnonReviewer4 · 2020-11-24
> > > > **Equation on the equality of KL diversion**
> > > >
> > > > Thank you authors! The paper is more readable now. Hopefully, it will be so to a broader audience through these changes.  Can the authors explain why the equality in the first line of equation 2 holds (i.e. equality regarding the subtraction of KL divergences)?

---

> > > > > ### Author Response · Authors · 2020-11-25
> > > > > **Proof of equation 2**
> > > > >
> > > > > We thank the reviewer for all the comments and suggestions. We have added a complete proof of eq. 2 in the Appendix (Lemma B.1) and we have added a reference to it in the main text.

---

### Official Review · AnonReviewer3 · 2020-10-29
**Interesting paper which needs further explanation of some experimental results & assumptions**

**Rating:** 7
**Confidence:** 3

**Review:**

Update: Thanks for the response. I agree with the other reviewers that the updates have improved the paper.

This paper presents a way of estimating the informativeness of a single training data point wrt a neural networks weights or it's output function.

Overall I think this paper is well written and interesting, but could benefit from links to other areas of the ML literature (detailed below) and further explanation of some puzzling experimental results.

The notion of unique information is very similar to the notion of strong relevancy in feature selection (in "Wrappers for Feature Subset Selection, Kohavi & John, Artificial Intelligence 1996). A feature is strongly relevant iff p(y|x,\omega) != p(y|\omega), that is if the presence of the feature changes the conditional distribution of the output conditioned on all the other features. In the same way a datapoint has unique information if the presence of it changes either the weight distribution or the output function. This much follows from the submitted paper. However the Kohavi & John paper also formalises a notion of weak relevance, which means that the presence of a feature changes the conditional distribution of the output given some subset of the other features. To capture all the information in a feature set you need all the strongly relevant features and some of the weakly relevant features (weak relevance means that the information appears multiple times in the feature set but it might not be in the strong relevant features so you need some of them to cover it). There is an information theoretic treatment of strong & weak relevance for feature selection in Brown et al "Conditional likelihood maximisation: a unifying framework for information theoretic feature selection", JMLR 2012 which may be of interest as it aligns with some of the presentation in the submitted paper. This notion of weak relevance would help formalise the discusson of least informative datapoints and the data summarisation paragraph in the experiments. The weakly relevant *datapoints* are those which describe the typical distribution of the dataset, a model needs some of them to properly capture the structure, but many of them provide the same information (and thus none will be uniquely informative) and that information may not appear in the uniquely informative datapoints.

The results in Table 1 are interesting, but there is little discussion of why the linearised formulation fails to capture the informative examples in the CNN trained from scratch. If the estimates diverge when the models become very different, then that is an important issue which should be discussed in more detail, so a reader can determine if the technique will be applicable to their use case.

The assumption that the SGD steady state covariance is unchanging implies to me that the example isn't very informative as otherwise it would change the loss landscape and thus change the covariance of SGD. Could the authors comment on the strength of this unchanging covariance assumption?

The informative-ness of datasources section seems to implicitly assume that the label distributions are balanced between the different sources, as otherwise the presence of rarely viewed labels could cause very large differences in the weights (which appears to cause trouble with the approximation technique given table 1). If this is required then it should be noted in the appropriate part of the discussion, otherwise could the authors comment on why it's not required?

Minor comments:

- The referencing style is inconsistent: some references say ICML, some give the volume in PMLR for that year's ICML, arxiv is cited in several different ways etc.

---

> ### Author Response · Authors · 2020-11-19
> **Response (part 1)**
>
> We thank the reviewer for the comments and suggestions.
>
> Feature selection and sample informativeness act on very different domains (one measures the average informativeness for the task variable of a group of features, the other measures the informativeness of a single sample for the final weights). However, we agree that they share several high-level similarities and similar notions. We thank the reviewer for the references, which we have added to our discussion in the revised paper.
>
> We also agree with the reviewer that typical samples which are redundant can be considered as weakly relevant (low-informative) datapoints, but are still needed by the network. That is, a sample can have low unique information when considered with all other examples, but be informative on its own. Indeed, in Fig. 2c we observe that removing 90% of the least informative examples performs worse than deleting 90% of samples at random. We hypothesize this is because all the typical samples are going to be removed, so that important information is lost. Random sampling, on the other hand, ensures that some typical samples will remain. We have added to Section 6 a small discussion on this topic.
>
> To better capture weak relevancy in our framework, we can consider the expected unique information an example has with respect to a random subset of examples, $\mathbb{E}_S I(z_i; w | S)$, with $i \not\in S$. Similar approach is employed in sample valuation using Data Shapley (Ghorbani and Zou [1], 2019). We added a discussion on this topic.
>
> >The results in Table 1 are interesting, but there is little discussion of why the linearised formulation fails to capture the informative examples in the CNN trained from scratch. If the estimates diverge when the models become very different, then that is an important issue which should be discussed in more detail, so a reader can determine if the technique will be applicable to their use case.
>
> There are two main reasons why linearization may not work well for the CNN trained from scratch. First, that the network (a ResNet-18 in our case) is not wide enough, and as we know linearization of randomly initialized networks becomes increasingly more accurate as we increase the width (Lee et al. [2], 2019). Second, when the network is randomly initialized, the weights have to change more to fit the task. This makes the Taylor approximation used for linearization less accurate. We have a small note about this in the first paragraph of page 7.
>
> [1] Ghorbani, Amirata, and James Zou. "Data shapley: Equitable valuation of data for machine learning." arXiv preprint arXiv:1904.02868 (2019).
> [2] Lee, Jaehoon, et al. "Wide neural networks of any depth evolve as linear models under gradient descent." Advances in neural information processing systems. 2019.

---

> ### Author Response · Authors · 2020-11-19
> **Response (part 2)**
>
> >The assumption that the SGD steady state covariance is unchanging implies to me that the example isn't very informative as otherwise it would change the loss landscape and thus change the covariance of SGD. Could the authors comment on the strength of this unchanging covariance assumption?
>
> The reviewer is right that an informative example changes the loss landscape and thus changes the covariance of SGD’s steady-state distribution. By assuming that the covariance doesn’t change, we ignore a part of the contribution of the example, that is, we only consider how much the example changes the global minimum of the loss function and not how much it changes the covariance of the steady state distribution of SGD. The benefit of this assumption is the simplicity of downstream definitions, and that it provides a simple connection of the effect of using SGD on the informativeness of samples. Assessing the strength of this assumption (which however we do not use in the rest of the paper and in the experiments), or completely eliminating it, is an interesting direction of research.
>
> Also, note that without the assumption, we can still use eq. 14 to compute the new steady-state covariance $\Sigma_{-i}$ obtained after removing one sample (removing one sample simply involves a rank-1 update of $H$ and $\Lambda$). Eq. 21 in the appendix then gives the correct (albeit slightly more complex) expression of SI for SGD.
>
>
> >The informative-ness of datasources section seems to implicitly assume that the label distributions are balanced between the different sources, as otherwise the presence of rarely viewed labels could cause very large differences in the weights (which appears to cause trouble with the approximation technique given table 1). If this is required then it should be noted in the appropriate part of the discussion, otherwise could the authors comment on why it's not required?
>
> The class imbalance and the quantity of data of a data source indeed affects the unique information of its samples. We believe this behavior is useful, for example to automatically discover which data sources have relatively fewer examples with respect to their complexity.In fact, we show something similar in the “Detecting under-sampled sub-classes” experiment, where we could alternatively interpret the two sub-classes as two different data sources.
>
> Note that we don’t expect rare examples to negatively affect the accuracy of the algorithm: Indeed, Table 1 suggests that the linear approximation is inaccurate when the final weights $w$ are too far from the initialization $w_0$ (as when training from scratch) or the network is not wide enough. While adding or removing a rare example may perturb $w$ more than other examples (i.e., $\|w - w_{-i}\|$ is large), we do not expect this perturbation to significantly affect the distance of $w$ from $w_0$ which is controlled instead by the most frequent examples.
>
> In the experiment in Fig. 2a (mixing MNIST and SVHN), even if not required, we had balanced classes in both data sources, to separate the effect of belonging to one or another data source from the effect of belonging to a rare class. Our goal was to show that images from one data source can be on average more informative than images from another data source.

---

### Author Response · Authors · 2020-11-19
**A revision is uploaded**

We thank all reviewers for their valuable comments and suggestions. We uploaded a revision of the manuscript that has the following changes.

- [R4] Added a clarification on the notations $A(S)$ and $A(w \mid S)$.
- [R1] Added missing information about architectures used in some experiments.
- [R3] Fixed inconsistencies in the references.
- [R3] Added a discussion on why removing more than 80% of the least informative examples degrades the performance more than removing the same number of random examples.
- [R1 & R2] Added new results providing evidence that sample information captures something inherent to the example (i.e., does not entirely depend on the training algorithm or the network architecture).
    - Computed correlations between functional sample information computed for 4 networks: ResNet-18, ResNet-34, ResNet-50, and DenseNet-121.
    - Did the MNIST vs SVHN experiment but with a different architecture (DenseNet-121 instead of ResNet-18) to show that the results stay qualitatively the same.
    - A new data summarization experiment, where the informativeness of examples are computed for one network, but another network is trained on data subsets. The results are qualitatively identical to those of the original experiment, confirming that information scores computed for one network can be useful for another network.
- [R2] Added an experiment that shows that adversarial examples are on average more informative. This hints that one can use the proposed information measures to detect adversarial examples.
- [R2] Added an experiment on sentiment analysis. This confirms that the proposed method can successfully work for non-visual modalities.
- [All reviewers] We added a discussion section that addresses some of the raised questions.
- [All reviewers] We did a new pass over the manuscript and fixed the typos.

---

### Decision · Program_Chairs · 2021-01-07
**Final Decision**

**Decision:**

Accept (Poster)

**Comment:**

This paper proposes methods to estimate how informative a single training data is wrt the weights and output of the neural network. All reviewers think this is an interesting problem and the proposed method is easy to implement. On the other hand, the reviewers also raise a few questions:
1.	There is a large body of work analyzing the informativeness of a feature wrt the model. The authors should compare their work to the feature importance analysis.
2.	The derived informativeness of a data depends not only on the network architecture, but also depends on the training algorithm, such as initialization and number of epochs. This makes the notion of data informativeness less general.
3.	The writing should be substantially improved.